# Computationally designed high specificity inhibitors delineate the roles of BCL2 family proteins in cancer

Stephanie Berger[1]\*, Erik Procko[2,3], Daciana Margineantu[4,5], Erinna F Lee[6,7,8,9,10], Betty W Shen[11], Alex Zelter[2], Daniel-Adriano Silva[2,12], Kusum Chawla[4,5], Marco J Herold[9,10], Jean-Marc Garnier[9,10], Richard Johnson[13], Michael J MacCoss[13], Guillaume Lessene[9,10,14], Trisha N Davis[2], Patrick S Stayton[1], Barry L Stoddard[11], W Douglas Fairlie[6,7,8,9,10], David M Hockenbery[4,5], David Baker[2,12,15]\*

[1]Department of Bioengineering, University of Washington, Seattle, United States; [2]Department of Biochemistry, University of Washington, Seattle, United States; [3]Department of Biochemistry, University of Illinois, Urbana, United States; [4]Clinical Research Division, Fred Hutchinson Cancer Research Center, Seattle, United States; [5]Human Biology Division, Fred Hutchinson Cancer Research Center, Seattle, United States; [6]Department of Chemistry and Physics, LaTrobe Institute for Molecular Science, Melbourne, Australia; [7]Olivia Newton-John Cancer Research Institute, Olivia Newton-John Cancer and Wellness Centre, Heidelberg, Australia; [8]School of Cancer Medicine, La Trobe University, Melbourne, Australia; [9]The Walter and Eliza Hall Institute of Medical Research, Parkville, Australia; [10]Department of Medical Biology, University of Melbourne, Parkville, Australia; [11]Basic Sciences Division, Fred Hutchinson Cancer Research Center, Seattle, United States; [12]Institute for Protein Design, University of Washington, Seattle, United States; [13]Department of Genome Sciences, University of Washington, Seattle, United States; [14]Department of Pharmacology and Therapeutics, University of Melbourne, Parkville, Australia; [15]Howard Hughes Medical Institute, University of Washington, Seattle, United States

\*For correspondence:
berger389@gmail.com (SB);
dabaker@uw.edu (DB)

**Competing interests:** The authors declare that no competing interests exist.

**Abstract** Many cancers overexpress one or more of the six human pro-survival BCL2 family proteins to evade apoptosis. To determine which BCL2 protein or proteins block apoptosis in different cancers, we computationally designed three-helix bundle protein inhibitors specific for each BCL2 pro-survival protein. Following in vitro optimization, each inhibitor binds its target with high picomolar to low nanomolar affinity and at least 300-fold specificity. Expression of the designed inhibitors in human cancer cell lines revealed unique dependencies on BCL2 proteins for survival which could not be inferred from other BCL2 profiling methods. Our results show that designed inhibitors can be generated for each member of a closely-knit protein family to probe the importance of specific protein-protein interactions in complex biological processes.

## Introduction

Programmed cell death is a tightly controlled process, involving both pro-survival and pro-apoptotic proteins that regulate permeability of the outer mitochondrial membrane. As cells enter apoptosis,

mitochondrial membrane permeability increases, releasing mitochondrial factors such as cytochrome *c* that initiate destructive protease cascades in the cytosol. The key regulators of mitochondrial outer membrane permeability are B cell lymphoma-2 (BCL2) family proteins which are categorized functionally by their effect on cell fate, and structurally by the presence of BCL2 homology (BH) motifs. Pro-apoptotic effector proteins Bak and Bax have four distinct BH motifs and homo-oligomerize upon activation to form pores in the mitochondrial outer membrane, committing the cell to apoptosis. Pro-survival homologs (six in humans: Bcl-2, Bcl-xL, Bcl-w, Mcl-1, Bfl-1 and Bcl-B) are structurally similar, but oppose apoptosis by binding and inhibiting Bak and Bax, as well as sequestering pro-apoptotic BH3-only proteins (BOPs). BOPs can also activate effectors directly through transient binding interactions (*Dai et al., 2011*; *Kim et al., 2009*; *Walensky et al., 2006*) or indirectly by binding pro-survival proteins and out-competing bound effectors (*Ku et al., 2011*; *Willis et al., 2007*) or other direct activator BOPs (*Kuwana et al., 2005*; *Letai et al., 2002*; *Figure 1*). Interactions between BCL2 members are mediated by an amphipathic, helical BH3 motif that recognizes a conserved hydrophobic cleft present in the effectors and pro-survival proteins. The balanced network of interactions between pro-apoptotic and pro-survival members can be tipped toward cell death by cellular stress signals that induce transcription (*Essafi et al., 2005*; *Nakano and Vousden, 2001*) or post-translational modification of BOPs (*Desagher et al., 2001*; *Fricker et al., 2010*; reviewed in *Shamas-Din et al., 2011*).

Pathology arises when apoptosis is dysregulated. Overexpression of one or more pro-survival homologs enables cancers to resist apoptosis, and different cancers have different profiles of pro-survival protein overexpression (*Kelly and Strasser, 2011*; *Placzek et al., 2010*). Small molecule and peptide therapeutics mimic BOPs by binding pro-survival proteins, inducing apoptosis by disrupting inhibition of Bak and Bax and limiting sequestration of BOPs. However, BH3-mimetics that non-specifically target multiple BCL2 proteins can cause harmful side effects by unnecessarily suppressing normal biological functions. For example, the small molecule ABT-737 (and related ABT-263) targeting Bcl-2, Bcl-xL and Bcl-w exhibits dose-limiting thrombocytopenia in treating Bcl-2-dependent chronic lymphocytic leukemia due to excessive inhibition of Bcl-xL, which has a role in platelet development (*Mason et al., 2007*; *Roberts et al., 2012*).

Delineation of the roles of pro-survival homologs in a given cancer, termed BCL2 profiling, aims to reveal which homolog or homologs a tailored treatment should target to maximize anti-cancer activity and minimize toxicity. BCL2 profiling using natural BOPs, BH3-mimicking peptides or small molecules is complicated by their low specificity (*Certo et al., 2006*; *Chen et al., 2005*; *DeBartolo et al., 2012*; *London et al., 2012*). Designed peptides and small molecules have achieved high affinity and excellent specificity for Bcl-2 (*Souers et al., 2013*), Bcl-xL (*Leverson et al., 2015a*), Mcl-1 (*Lee et al., 2008*; *Foight et al., 2014*; *Leverson et al., 2015b*), and Bfl-1 (*Dutta et al., 2013*), and highly specific small molecule inhibitors of Bcl-2 and Bcl-xL (ABT-199 and A-1155463) have defined the dependency of ABT-263-sensitive cancer cell lines on Bcl-2, Bcl-xL or both (*Leverson et al., 2015a*). However, there are currently no highly specific inhibitors for Bcl-w and Bcl-B, and hence general mechanistic aspects of apoptotic regulation remain unclear. Here we describe the computational design and experimental characterization of specific, high affinity protein inhibitors for all six pro-survival BCL2 homologs (*Figure 1—figure supplement 1*). The inhibitors exhibit high specificity in engineered cell lines, and in defined combinations they induce apoptosis in representative cancer cell lines. This comprehensive set of molecular probes should be useful to elucidate the molecular mechanisms of mitochondrial apoptotic pathways, determine BCL2 profiles of individual cancers, and provide a superior guide for tailored therapies.

## Results

### Computational design of BCL2 binding proteins

We recently described a de novo designed protein inhibitor of BHRF1, an Epstein-Barr viral BCL2 homolog. The three helix bundle protein, called BINDI, is complementary to the canonical BH3-binding groove of BHRF1. BINDI consists of a central BH3-like motif and two additional helices that both stabilize the BH3-motif and provide extra contacts for high affinity and specific binding (PDB 4OYD; *Procko et al., 2014*). Pro-survival BCL2 homologs share similar sequences (40–60% similarity between any two) and structures (approximately 3 Å RMSD), and hence achieving specific binding

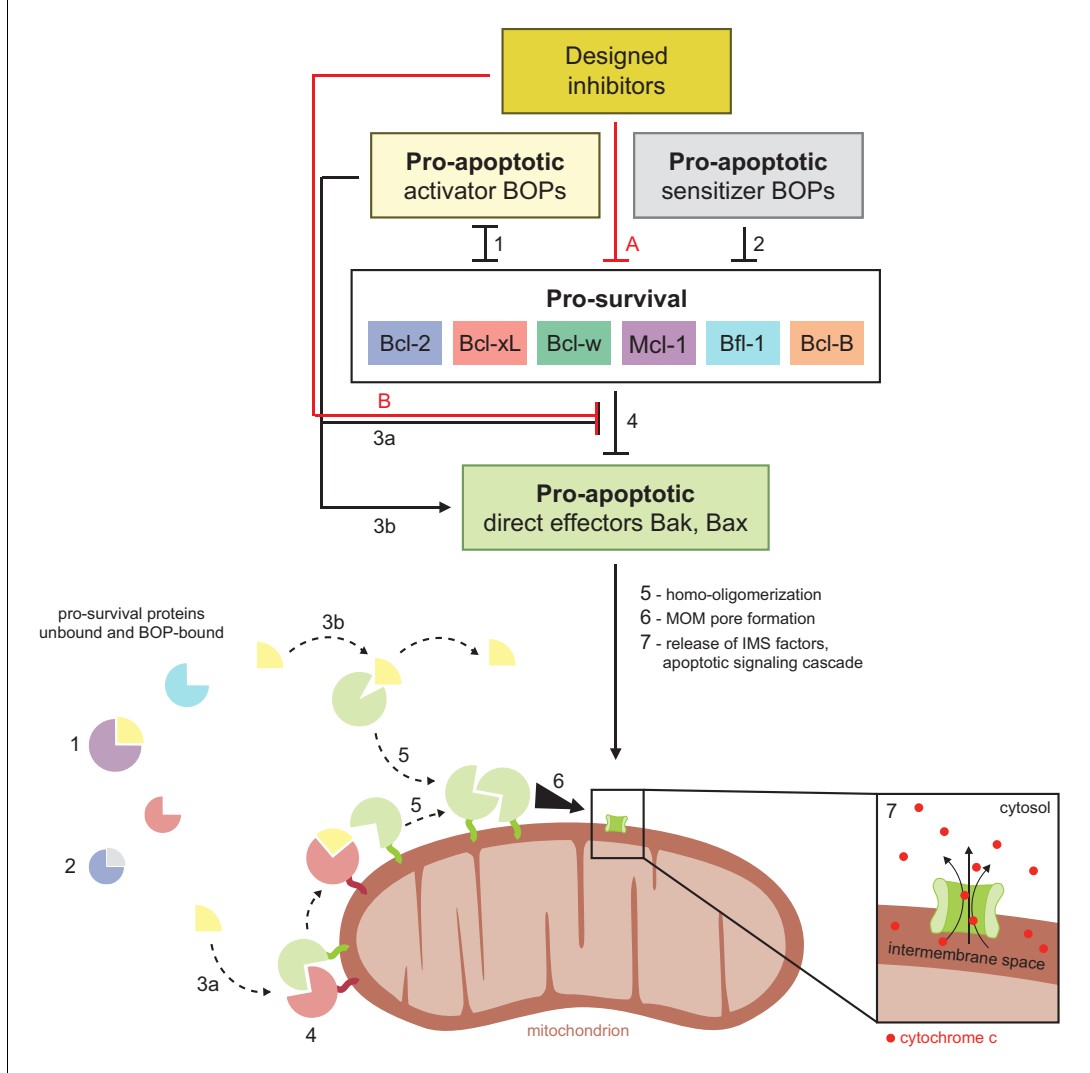

**Figure 1.** Schematic of BCL2 family interactions. BCL2 proteins are categorized by their net effect on cell fate and the presence of shared structural domains. BH3-only proteins (BOPs) are sequestered by pro-survival homologs (labels **1** and **2**), and some BOPs may activate the direct effectors Bak and Bax by disrupting their inhibition by pro-survival proteins (**3a**) and/or promoting their homo-oligomerization (**3b**). Pro-survival proteins, which are typically overexpressed in cancer, bind and inhibit Bak and Bax (**4**), which would otherwise homo-oligomerize upon activation (**5**) and form pores in the mitochondrial outer membrane (MOM; **6**). MOM permeabilization allows the release of cytochrome c and other factors from the intermembrane space (IMS) and thus initiates the apoptotic signaling cascade (**7**). Designed inhibitors have a net pro-apoptotic effect by binding pro-survival proteins, which may both limit sequestration of BOPs (**A**) and disrupt inhibition of Bak and Bax (**B**).

The following figure supplement is available for figure 1:

**Figure supplement 1.** Design strategy.

for each one is a challenging problem. We hypothesized that the expanded binding interface of the BINDI scaffold could enable design of specificity by contacting regions where BCL2 homolog sequences differ both within and outside of the conserved BH3 binding cleft (*Figure 2*).

The BINDI scaffold was docked into the hydrophobic binding cavities of crystal structures of the six pro-survival homologs bound to various ligands (*Supplementary file 1A*). If the target structure included a bound BH3 motif, this was used to structurally align the BH3-equivalent residues of BINDI in the binding groove. If the target structure was bound to an unnatural ligand, such as a small molecule or α/β-foldamer, the model of the pro-survival homolog was first aligned to an alternative

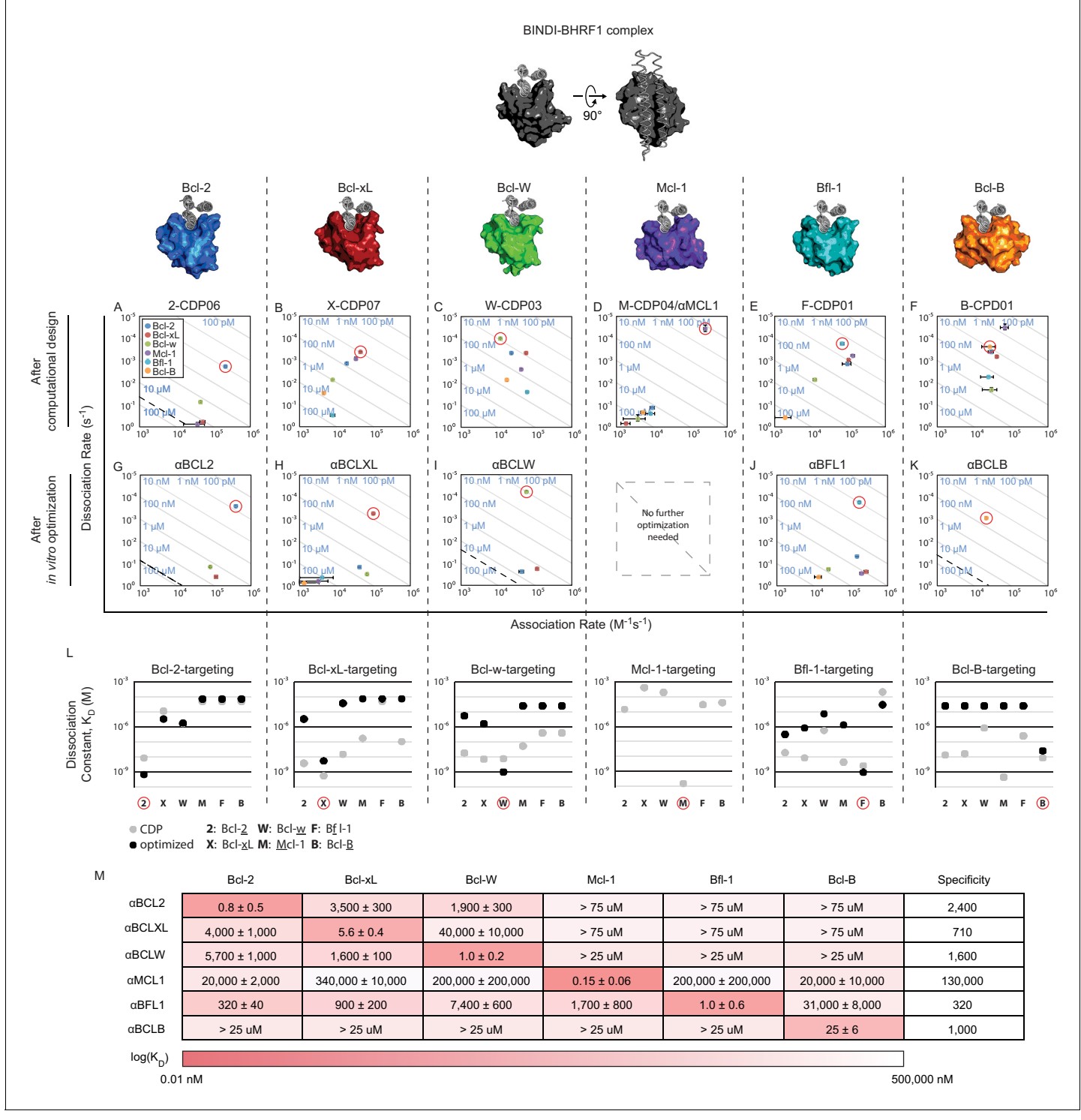

**Figure 2.** Design of specific inhibitors for each of the six human pro-survival BCL2 homologs. On and off rates were determined by BLI with multiple-concentration binding titrations for each computationally designed protein (**A**–**F**) and optimized variants (**G**–**K**; mean ± SD; n = 3). On-target interactions are indicated with red circles. Diagonal lines represent dissociation constants ($K_D$) as labeled. Dashed lines indicate affinities at which binding signals were too weak to be accurately measured; dissociation constants for interactions not plotted are assumed to be greater than these thresholds. (**L**) $K_D$ values for computational designs before (gray) and after optimization (black). (**M**) $K_D$ values for final optimized inhibitors (mean ± SD; n = 3).

The following source data and figure supplement are available for figure 2:

*Figure 2 continued on next page*

*Figure 2 continued*

**Source data 1.** Source data relating to *Figure 2A–M* and *Figure 2—figure supplement 1H*.
**Source data 2.** Source data relating to *Figure 2—figure supplement 1I*.
**Figure supplement 1.** Computational design and screening methods.

structure bound to a helical BH3 motif, which then served as a guide for structural alignment of BINDI. One docked model was generated for each crystal structure. Key interfacial residues were transferred to the BINDI scaffold (*Correia et al., 2010*), borrowing side chains from each crystal structure's bound peptide ligand, and informed by peptide SPOT array data (*DeBartolo et al., 2012*) and the sequences of selective BOPs and BH3-mimetic peptides (*Chen et al., 2005*; *Dutta et al., 2010*; *Supplementary file 1A*).

Following docking and side chain grafting, ROSETTA Monte Carlo sequence design calculations were carried out on BINDI residue positions within 8 Å of the target interface to minimize the energy of the bound complex (*Leaver-Fay et al., 2011*). Grafted residues and protein backbone conformations were kept fixed. Side chain rotamers of the target BCL2 homolog were allowed to sample alternative conformations compatible with the redesigned interface. In a second round of design calculations, the designable interface was expanded to include BINDI residues within 12 Å of the target, followed by rigid-body minimization. Five to 10 designs were generated for each initial docked configuration, and those with the most favorable binding energy, smallest number of buried polar atoms, and greatest shape complementarity to the target's surface were selected.

Genes encoding the selected designs were synthesized, and nearly all the proteins were expressed and soluble in *E. coli* (summary in *Supplementary file 1A*; sequences in *Supplementary file 1B*). The purified proteins were screened with single-concentration biolayer interferometry (BLI; *Figure 2—figure supplement 1F*) to qualitatively assess affinity and specificity for the target BCL2 protein. The affinities of the most specific designs were quantitatively determined using multiple-concentration BLI (*Figure 2—figure supplement 1G*). 2-CDP06 (for Bcl-2-targeting Computationally Designed Protein), X-CDP07 (Bcl-xL), M-CDP04 (Mcl-1), and F-CDP01 (Bfl-1) bound their intended targets with highest affinity, while the affinity of B-CDP01 to its intended target Bcl-B was second only to Mcl-1 (*Figure 2* and *Figure 2—figure supplement 1H*).

Initial Bcl-w-targeting designs, however, did not bind Bcl-w or any other BCL2 protein, likely because the designs were based on the crystal structure of Bcl-w bound to a ligand that is not BH3-like (PDB 4K5A), unlike successful designs that were based on BH3-liganded structures (*Supplementary file 1A*). Therefore, we generated helix-bound Bcl-w models by threading the Bcl-w sequence onto high-resolution structures of other homologs bound to BH3 peptides and sampled alternative superpositions of the BINDI scaffold onto the modeled BH3 peptide (*Figure 2—figure supplement 1C*). Each docked conformation was then designed as described above, and 36 sequences passing design filters were pooled and expressed on the yeast cell surface as fusions with Aga2p. The yeast library was sorted by fluorescence-activated cell sorting (FACS) for binding to biotinylated Bcl-w in the presence of the other BCL2 pro-survival homologs as unlabeled competitors; this enriched designs with high affinity and specificity for Bcl-w. Enriched designs were expressed in *E. coli* and screened by BLI. Design W-CDP03 was the most specific, binding Bcl-w with nanomolar affinity and moderate specificity (*Figure 2C*, *Figure 2—figure supplement 1H*). Notably, the location of the BH3-like motif in W-CDP03 is shifted by one α-helical turn relative to BINDI, perhaps to better accommodate the Bcl-w surface (*Figure 2—figure supplement 1D and E*).

## The αMCL1•Mcl-1 crystal structure is very similar to the design model

The computational design calculations succeeded in generating proteins that bound to each of the six human BCL2 homologs with nanomolar affinity and at least partial specificity. One design, M-CDP04 (subsequently called αMCL1, or anti-Mcl-1), was highly specific for Mcl-1 and bound with picomolar affinity. Cross-linking studies of αMCL1 with Mcl-1 were consistent with the designed binding interactions, supporting the structural model at low resolution (*Figure 3—figure supplement 1*, *Supplementary file 1D*).

The crystal structure of the αMCL1•Mcl-1 complex at 2.75 Å resolution reveals how high affinity and specificity were achieved (*Figure 3*, *Supplementary file 1C*). When Mcl-1 in the design model is superimposed on Mcl-1 in the crystal structure, αMCL1 crystal and design models closely align, highlighting the accuracy of our design calculations (2.1 Å average RMSD among the six separate complexes observed in the asymmetric unit; the N-terminal end of αMCL1 in the crystal structure lying slightly closer to Mcl-1 than in the design). The high specificity and affintiy result from many precisely positioned designed sidechains.

Native BH3 motifs interact with pro-survival homologs via defined hotspot residues on five consecutive turns of the BH3 helix, denoted h0 through h4 (*Figure 4A*). The BH3-mimetic helix 2 of αMCL1 has three additional helix turns beyond h0 and h4 that have side chains close enough to interact with Mcl-1. These extra contacts, combined with those made by the peripheral helices, expand the classic BH3 interface by 534 Å$^2$ (*Figure 3B*). While many residues in the αMCL1 BH3-mimetic helix were borrowed from pan-specific Bim (*Supplementary file 1A*), designed residues at the expanded interface provide tailored complementarity with Mcl-1 for improved affinity and specificity (*Figure 3C–F*).

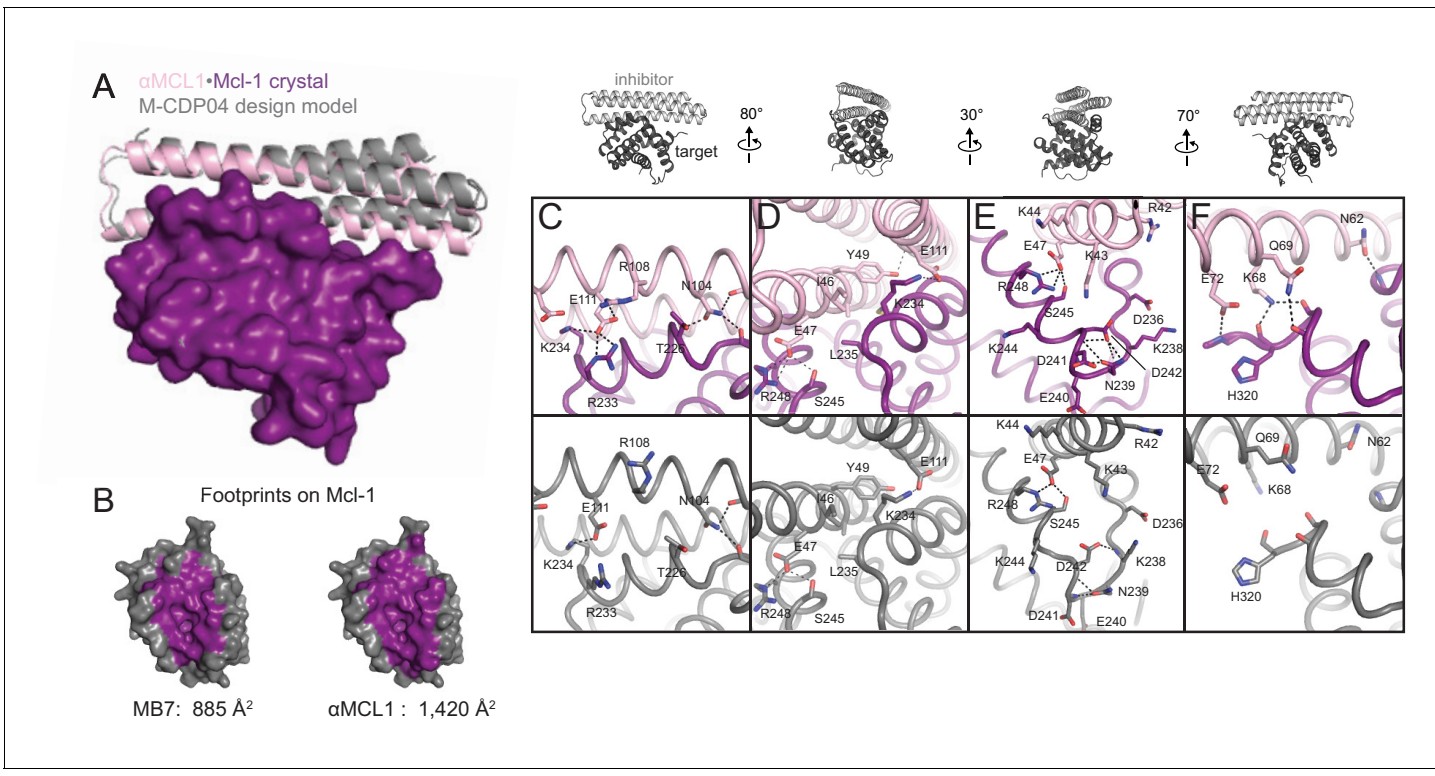

**Figure 3.** The crystal structure of αMCL1•Mcl-1 is very close to the design model. (**A**) Alignment of the computational design model M-CDP04 (gray cartoon) and crystal structure (Mcl-1, purple surface; αMCL1, pink cartoon). (**B**) Buried contact surfaces on Mcl-1 bound to a BH3-like motif (designed peptide MB7; PDB 3KZ0) and αMCL1. (**C–F**) Comparison of crystal structure (top panels) with the design model (bottom panels) highlights accuracy of design and shows how high specificity was achieved. (**C**) αMCL1 computationally designed residues E111, R108 and N104 complement nearby Mcl-1 residues. (**D**) αMCL1 residue 46 was redesigned from glutamate (BINDI scaffold) to isoleucine to accommodate the hydrophobic Mcl-1 binding pocket. (**E**) Designed residues R42, K43 and K44 promote long-range electrostatic complementarity to the negatively-charged loop region of Mcl-1. αMCL1 residue E47 (borrowed from Bim) makes ionic interactions with Mcl-1 residues S245 and R248. (**F**) Designed residues K68, Q69, and E72, and Bim residue N62, make polar interactions with the Mcl-1 backbone. Though the design model does not place αMCL1 near enough to Mcl-1 to make these interactions, the design calculations selected residues with long-range electrostatic complementarity.

The following figure supplement is available for figure 3:

**Figure supplement 1.** Structural analysis of the αMCL1•Mcl-1 complex via lysine-specific chemical cross-linking.

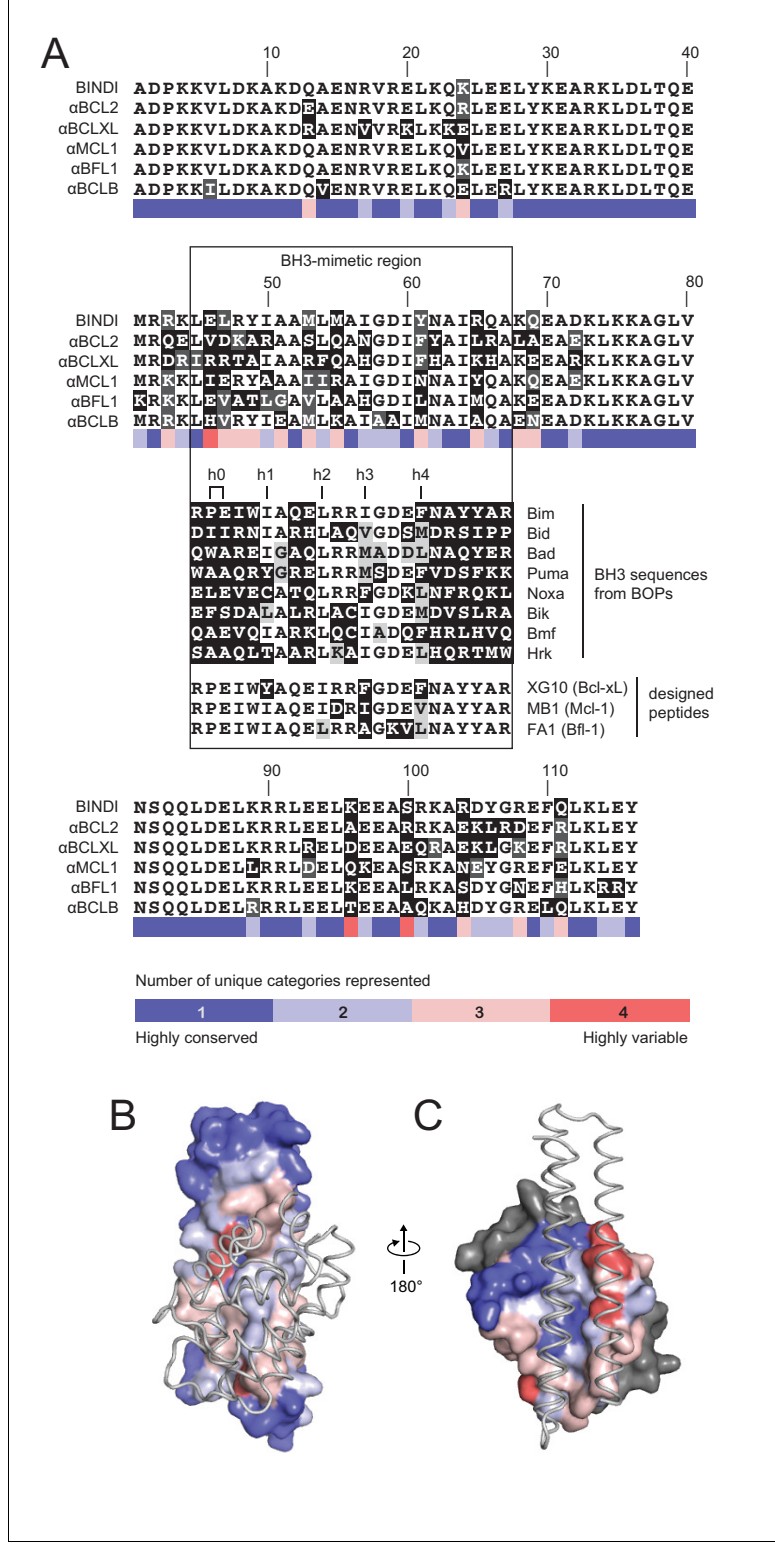

**Figure 4.** Comparison of design sequences with BH3-mimetic peptides and natural BH3 motifs. (**A**) Sequences of optimized inhibitors are aligned, excluding αBCLW, which binds to Bcl-w using a shifted interaction surface. The BH3-mimetic region of designed inhibitors is compared to natural BH3 sequences and synthetic peptides designed for indicated specificities. Non-consensus residues are shaded gray if similar to consensus and black if different. (**B**) Conservation was assessed by counting the number of unique categories of amino acids (polar, charged, etc.) represented across each position. Conservation scores were mapped onto each position of BINDI

*Figure 4 continued on next page*

*Figure 4 continued*

(surface) bound to BHRF1 (gray ribbon; PDB 4OYD). (**C**) Conservation scores from a sequence alignment of BCL2 proteins are mapped to BHRF1 (surface) bound to BINDI (gray ribbon). The designed proteins differ considerably from BOPs and previously designed peptides and contain many additional specificity-enhancing residues outside the BH3 region.

## Affinity and specificity maturation

To improve the affinity and specificity of the designed inhibitors targeting other BCL2 homologs, the genes for 2-CDP06, X-CDP07, W-CDP03, F-CDP01 and B-CDP01 were diversified by site-directed saturation mutagenesis (SSM). Each codon was mutagenized to NNK (N is A, G, C or T; K is G or T) by overlap PCR (*Procko et al., 2013*), producing a library comprising all possible single amino acid substitutions. Each library was screened by yeast display for specific binding to labeled target homolog in the presence of unlabeled competitors (sort conditions in *Supplementary file 1E*). DNA from the naïve and post-sort libraries was extracted and deep sequenced.

The enrichment or depletion of each sequence variant in the selected versus unselected pools is a measure of the variant's fitness with respect to affinity and/or specificity toward the target homolog (*Figure 5—figure supplement 1A*). Enriching mutations were found on the central BH3-mimicking helix and at positions on the peripheral helices that contact the target. To assess the accuracy of the computational design in identifying optimal amino acids at the interface, we calculated the deviation of each designed residue's enrichment ratio from the maximum enrichment ratio at that position. For all CDPs, nearly all designed residues have enrichment ratios very close to the maximum (*Figure 5A*, *Figure 5—figure supplement 1B*), on average deviating by 2.2 (2.2-fold worse enrichment) while the average deviation per position is 4.1 (*Figure 5B*).

To experimentally evaluate the contribution of computational design, we carried out control evolution experiments starting from a single, partially-specific Mcl-1-targeting design aiming for specificity toward each of the other pro-survival BCL2 proteins. An SSM library based on M-CDP02 was

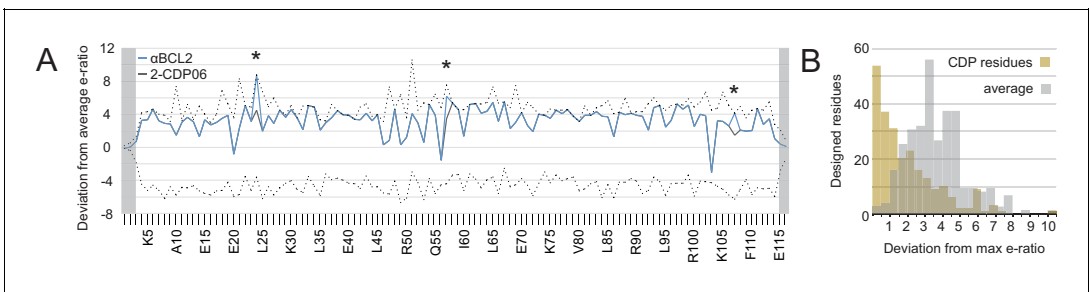

**Figure 5.** Analysis of computational design success. (**A**) Deep sequencing analysis of the naïve and sorted 2-CDP06 SSM library enabled quantitative analysis of the fitness of each single amino acid substitution for specificity and affinity toward Bcl-2. Per position, the enrichment ratio (abbreviated e-ratio; a fitness score) of each 2-CDP06 residue (gray) was compared to the average value for all 20 amino acids (normalized to zero). Maximum deviations from the average are represented by dashed lines, positive values indicate the best score and negative the worst. SSM-guided mutations from 2-CDP06 to αBCL2 (blue) are starred. Gray shading indicates positions with insufficient sequencing data. (**B**) Deviation from maximum e-ratio was calculated for each designable residue of the five mutagenized CDPs, pooled, and the distribution of deviations plotted (gold; full SSM heatmaps in *Figure 5—figure supplement 1*); distribution of average deviations from maximum for each designable residue is shown in gray.

The following source data and figure supplements are available for figure 5:

**Source data 1.** Source data relating to *Figure 5* and *Figure 5—figure supplement 1*.

**Figure supplement 1.** Sequence analysis of SSM libraries.

**Figure supplement 2.** Computational docking calculations: CDPs.

**Figure supplement 3.** Computational docking calculations: optimized inhibitors.

sorted as described above (sort conditions in *Supplementary file 1E*). Mutations that enhance the affinity of M-CDP02 for BCL2 members other than Mcl-1 include prolines in the first and third helical segments, substitutions of apolar to polar amino acids in the hydrophobic core, and premature stop codons in the third helix. These mutations likely cause unfolding of the helix bundle and expose the Bim-BH3-like motif in the second helix, thus converting a protein that binds Mcl-1 with high affinity and partial specificity to a pan-specific high-affinity binder similar to the Bim-BH3 motif (*Figure 5—figure supplement 1C*). In contrast, none of these destabilizing mutations were enriched during the evolution of the individual computational designs explicitly targeting each BCL2 homolog. Thus, using our experimental approach, computational design is necessary to provide partially-specific starting points for evolution which are superior to a non-specific construct.

For X-CDP07, W-CDP03, F-CDP01 and B-CDP01, combinatorial libraries were constructed containing the mutations that produced the greatest increase in specificity (highlighted in *Figure 5—figure supplement 1A*; *Supplementary file 1F*), and sorted by FACS for multiple rounds under increasingly stringent conditions (*Supplementary file 1E*). Each library converged on a small number of enriched combinatorial mutants (ECMs), which were screened by BLI. We anticipated that only a small number of substitutions in the moderately-specific 2-CDP06 design would be necessary to achieve high specificity for Bcl-2. Thus, in lieu of generating a combinatorial library, single amino acid mutants were screened with BLI, and three mutations improving both specificity and affinity were combined in αBCL2 (*Figure 2G*, *Supplementary file 1F*).

While X-ECM04 and W-ECM01 (hereafter called αBCLXL and αBCLW) have high affinity and excellent specificity (*Figure 2H–I*), F-ECM04 and B-ECM01 exhibited less than 100-fold specificity for their targets. These sequences were therefore diversified by error-prone PCR, evolved and screened as previously (*Supplementary file 1E*). Three additional specificity-enhancing mutations were identified per construct and combined in the final variants αBFL1 and αBCLB (*Figure 2J–K*, *Supplementary file 1F*). Overall, the optimized designs exhibit slight to moderate decreases in stability compared to their predecessors based on chemical denaturation, but unfolding remains cooperative (*Figure 2—figure supplement 1I*), suggesting a well-packed core.

We carried out computational docking experiments on partially specific CDPs and optimized variants to assess the robustness of our computational protocol (*Figure 5—figure supplements 2* and *3*). Each CDP and optimized inhibitor was docked into the canonical binding groove of each BCL2 homolog, and thousands of docked configurations were sampled both locally (low RMSD to input configuration) and globally (entire protein surface). Overall, both the partially-specific CDPs and optimized, specific inhibitors exhibit more favorable absolute binding energy (local minimum ddG) and relative binding energy (local minimum versus global minimum ddG) when docked to on-target homologs compared to off-target homologs. These calculations resemble trends in the experimental binding data, but they do not discriminate between the highly specific, optimized inhibitors and partially specific precursors. Thus, while adding computational docking or multi-state design to computationally select against off-target homologs to our design protocol may improve the initial success rate of achieving high affinity and at least partially specific binding, the resolution of these calculations limits discrimination between variants with low versus high specificity.

## Determinants of specificity

The crystal structure of the αBCL2•Bcl-2 complex at 2.1 Å resolution together with the αMCL1•Mcl-1 complex described above illuminate the structural basis for affinity and specificity achieved by both computational design and evolution. The sequence variability of the designed proteins complements that of the BCL2 proteins across the interface, indicating that the designed proteins gain specificity by taking advantage of regions where BCL2 homologs differ (*Figure 4B–C*). Mutations that enhanced specificity localize to three regions: the interface periphery, the hydrophobic core, and the BH3-like region.

Many mutations at the interface periphery change surface electrostatic potential to improve charge complementarity with the target or oppose interactions with off-target BCL2 proteins. For example, designed negatively charged residue E111 of αMCL1 complements a positively charged region of Mcl-1 and opposes negatively charged analogous regions of Bcl-2, Bcl-xL and Bfl-1. αBCL2, αBCLXL and αBFL1 each have designed (αBCL2) or evolved (αBCLXL, αBFL1) positively charged side chains at position 111, which likewise complement on-target binding and oppose binding to Mcl-1 (αMCL1•Mcl-1 and αBCL2•Bcl-2 crystal structures shown in *Figure 6A*; Bcl-xL and Bfl-1

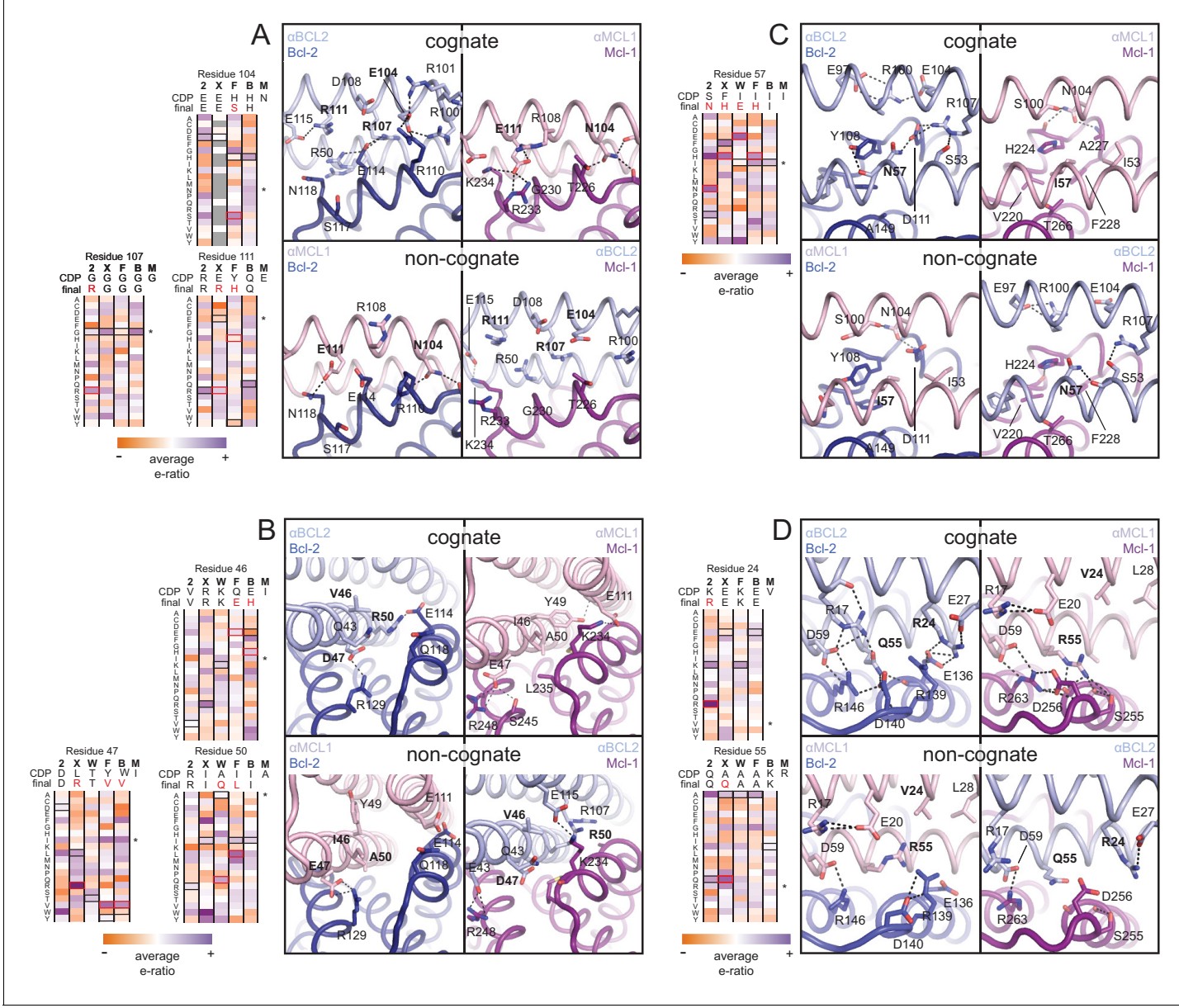

**Figure 6.** Determinants of binding specificity. αMCL1•Mcl-1 and αBCL2•Bcl-2 crystal structures (upper panels, high complementarity) and non-cognate binding pairs modeled in Rosetta (lower panels, poor complementarity) were aligned. For select positions on the three-helix bundle scaffold, normalized enrichment of each mutant (indicated by amino acid code) toward specific binding to each homolog (indicated at the top of each column) are shown for comparison Black outlines indicate the identity of the homolog-specific CDP, and red outlines indicate the identity of the homolog-specific optimized inhibitor (if different from CDP). Stars indicate the identity of M-CDP04/αMCL1 (no in vitro evolution required, and thus no deep sequencing data available). Gray fill indicates positions with insufficient sequencing data. Analogous αBCLW residues were included for helix 2 (sequence shifted + 4 relative to others). (A) Designed αBCL2 residues E104 and R111 and αMCL1 N104 and E111 illustrate computational design success. Each contributes polar contact(s) with its target homolog, and deep sequencing data show these residues deplete binding toward one or more competitor homologs to improve specificity. αMCL1 E111 opposes Bcl-2 E114. SSM-guided αBCL2 mutation G107R contributes additional polar contacts with Bcl-2. (B) Designed αBCL2 residue R50 is tolerated by a more spacious Bcl-2 binding pocket and interacts with Bcl-2 E114. Designed αBCL2 residue D47 is partially satisfied by Bcl-2 R129. Both αBCL2 R50 and D47 fit poorly in the more hydrophobic analogous region of Mcl-1. (C) Evolved αBCL2 residue N57 introduces polar atoms in the hydrophobic interface but is partially satisfied by Bcl-2 D111. (D) Evolved αBCL2 residue R24 and designed Q55 make polar contacts with Bcl-2. αMCL1 R55, borrowed from Bim, caps an Mcl-1 helix and opposes Bcl-2 residue R139.

The following figure supplement is available for figure 6:

**Figure supplement 1.** The crystal structure of the αBCL2•Bcl-2 complex.

comparison using structural alignment of existing models). Additional examples of designed and evolved electrostatic complementarity are illustrated in *Figure 6B–D*.

Conservative mutations in the hydrophobic core may improve core packing or alter the backbone conformation for enhanced complementarity to the target surface. For example, the binding mode of αBCL2 in the hydrophobic cleft of Bcl-2 differs significantly between the crystal structure and backbone-constrained design model; after Bcl-2 alignment, Cα backbones of the αBCL2 crystal and design models deviate by 4.0 Å RMSD (average amongst the two complexes observed in the asymmetric unit; *Figure 6—figure supplement 1A–C*). The SSM-guided mutation of 2-CDP06 core residue G107R is likely responsible, requiring the first and third helices of αBCL2 to shift relative to the BH3-mimetic helix and positioning the third helix much further from Bcl-2 than the αMCL1•Mcl-1 binding mode (*Figure 6A*). The αBCL2 binding mode enables electrostatic interactions between αBCL2 R107 and Bcl-2 residues D111 and E114.

Mutations within the hydrophobic center of the interface, formed by the BH3-like region of the designs, were generally conservative, but occasionally included substitutions of hydrophobic to polar residues. In particular, the position analogous to a conserved isoleucine within natural BH3 motifs (h3 in *Figure 4A*) is mutated to a polar residue in αBCL2 (N57), αBCLXL (H57), αBCLW (E61) and αBFL1 (H57). Mutation of this residue was not allowed during the design of αMCL1 or the design and evolution of αBCLB, which therefore both preserve the isoleucine hotspot. The αBCL2•Bcl-2 crystal structure reveals that Bcl-2 residue D111 makes a hydrogen bond with αBCL2 N57, satisfying a polar atom that is likely buried in the interface when binding other homologs (*Figure 6C*). Specificity appears to be achieved in part by introducing a small number of mutations that universally reduce binding affinity but improve specificity at the interface center, like αBCL2 N57 which can be tolerated by Bcl-2 but likely reduces binding to other homologs, coupled with many specific, affinity-enhancing mutations at the interface periphery.

Engineered BH3-mimetic peptides span residues analogous to the BH3-like core interface of the designed inhibitors. The specificity of small peptides thus depends on mutations within this limited region. Like αMCL1, αBCL2 expands the classic BH3 interface by 452 Å$^2$ (*Figure 6—figure supplement 1D*). While the designed proteins share some specificity-enhancing residues with designed peptides (*Dutta et al., 2010*, *2013*), they also conserve non-specific residues at these positions; for example, aspartate at position h2 + 1 of the MB1 peptide is thought to confer specificity to Mcl-1, but αMCL1 retains arginine as in pan-specific BOP Bim (*Figure 4D*). Further, several positions that contribute to the specificity of designed peptides and some BOPs are restricted in the designed proteins to conserved hydrophobic residues as they fall within the helix bundle's core (h1 + 2, h2 + 2, and h3 + 3; *Figure 4A*). Our design strategy achieves specificity by employing a lower-affinity central interface and designing additional interactions over the expanded target-inhibitor interface.

## Validation of binding specificity and mechanism in engineered cell lines

We investigated the BCL2 binding profiles and mechanism of action of the optimized inhibitors in mammalian cells, employing a suite of engineered mouse embryonic fibroblasts (MEFs). We tested whether our inhibitors could selectively induce a hallmark of apoptosis by monitoring cytochrome *c* release from mitochondria into the cytosol of MEFs with engineered dependence on a single prosurvival BCL2 homolog. Strikingly, permeabilized MEFs treated with each designed inhibitor induced cytochrome *c* release only in the cell line dependent on the corresponding target BCL2 protein. No cytochrome *c* release was observed in *Bak$^{-/-}$Bax$^{-/-}$* cells, confirming that mitochondrial outer membrane permeability following inhibitor treatment occurs specifically via the BCL2-regulated intrinsic pathway, as expected (*Figure 7A*).

To further validate binding specificity we examined the effect of a subset of inhibitors (αMCL1 and αBFL1) on long-term (i.e. seven day) colony survival in MEFs engineered to inducibly express each inhibitor. Consistent with binding profiles and cytochrome *c* release data, large effects were only seen with αMCL1 in the Mcl-1-dependent line, causing a 90 ± 11% decrease in survival, and with αBFL1 in the Bfl-1-dependent line, causing a 85 ± 6% decrease in survival (*Figure 7—figure supplement 1A*). Minimal effects on cell survival were observed in lines expressing non-cognate prosurvival proteins. These data validate the specificity of the designed proteins and their capacity to functionally engage BCL2 family members in a cellular milieu.

While engineered MEFs provided an excellent model system to study our designed proteins, we sought further mechanistic validation in a context relevant to their primary application: probing

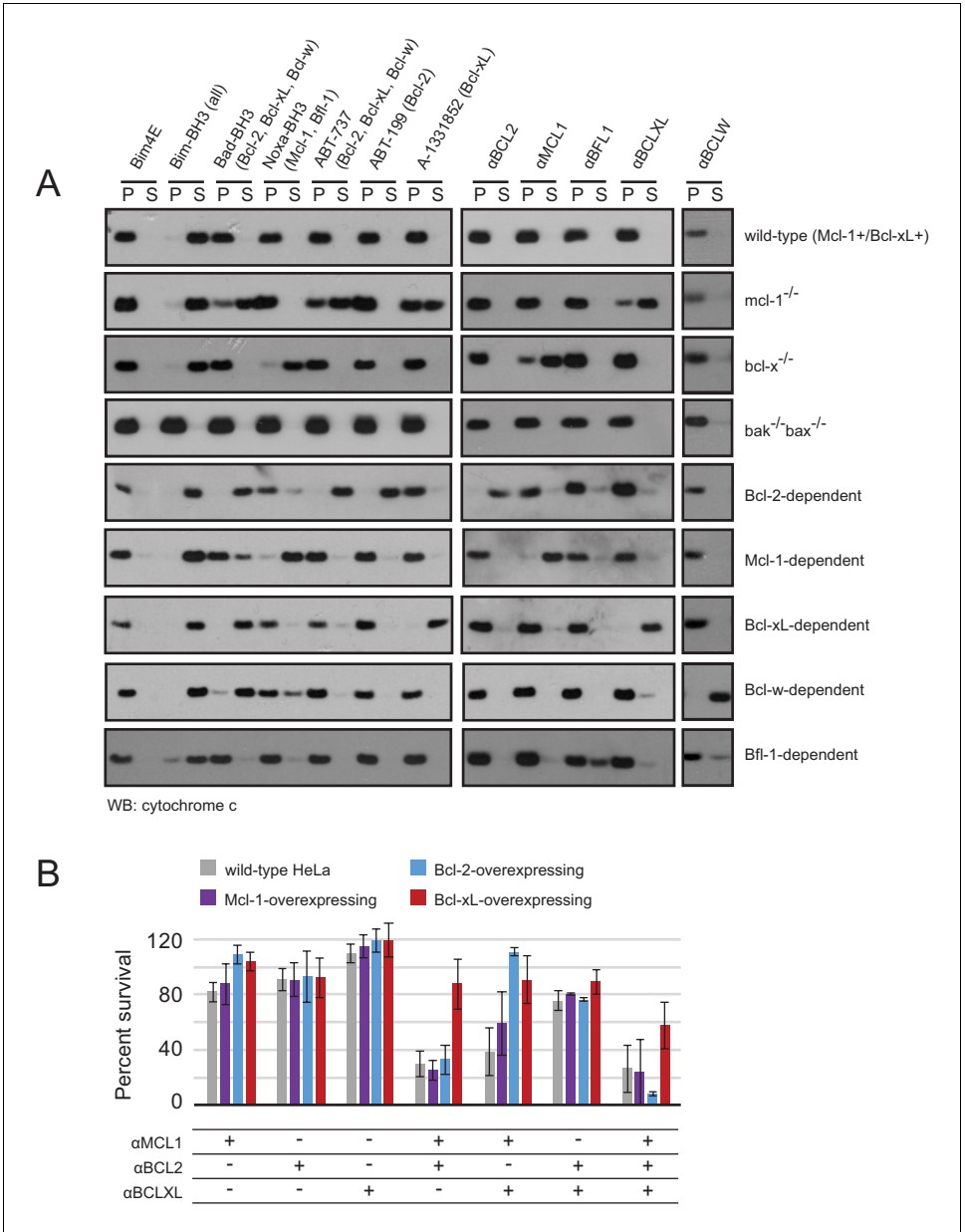

**Figure 7.** Designed inhibitors induce apoptosis in vitro by engaging the BH3-binding grooves of specific pro-survival homologs. (**A**) Western blot for cytochrome *c* in pelleted (P) and soluble (S) fractions of engineered MEFs after permeabilization and treatment with 10 mM BCL2 inhibitors. Bim-BH3, which binds all pro-survival homologs, is a positive control. Bim-BH3 peptide with four mutations to glutamate at interface residues (Bim4E) is a negative control. BOPs Bad and Noxa, and small molecule drugs tested have the indicated binding specificities in parentheses. (**B**) HeLa cells were transduced with constructs for designed inhibitor expression, and viability was assayed after 72 hr (mean ± SD; n = 2 for Bcl-2+ double and triple combinations, n = 3 for all others).

The following source data and figure supplement are available for figure 7:

**Source data 1.** Source data relating to *Figure 7B* and *Figure 7—figure supplement 1A*.

**Figure supplement 1.** Long-term MEF survival and HeLa co-immunoprecipitation studies.

BCL2 family interactions and generating functional BCL2 dependency profiles in cancer. A representative cancer cell line (HeLa) was engineered to overexpress Mcl-1, Bcl-2 or Bcl-xL, and we assayed the activity of the designed inhibitors in each setting (*Figure 7B*). Previous studies revealed that HeLa cells are resistant to the expression of Noxa (which targets Mcl-1 and Bfl-1) and ABT-737 (Bcl-2 and Bcl-xL) independently, but are potently killed with the combination of Noxa with ABT-737 (*van Delft et al., 2006*). Likewise, single designed inhibitors had little effect on survival. More substantial cell death was induced by combinations of αMCL1 with αBCL2 (29 ± 9% survival) and αMCL1 with αBCLXL (38 ± 17%) than αBCL2 with αBCLXL (75 ± 7%). These data, and similar results in Mcl-1-overexpressing (Mcl-1+) HeLa cells, suggest that Mcl-1 plays a more crucial role in wild-type HeLa survival than Bcl-2 or Bcl-xL.

Compared to wild-type and Mcl-1+ HeLa cells, Bcl-xL-overexpressing (Bcl-xL+) cells are more resistant to the combination of αMCL1 with αBCL2, and likewise, Bcl-2-overexpressing (Bcl-2+) cells are more resistant to the combination of αMCL1 with αBCLXL. Thus, increased expression of a given BCL2 protein can compensate for the inhibition of others. The triple combination of αMCL1, αBCL2, and αBCLXL had greater efficacy than double combinations, indicating a contribution of each pro-survival protein to basal survival. Bcl-xL+ cells were generally more resistant than all other cell lines; the inability to completely inhibit Bcl-xL's survival function in Bcl-xL+ cells suggests that in this context, Bcl-xL may interact with proteins that are not displaced efficiently by αBCLXL.

To investigate potential mechanisms underlying these results, we assessed the binding profile of a representative BOP, Bim, to pro-survival homologs with co-immunoprecipitation (co-IP) experiments in wild-type and over-expressing cell lines, with and without added αMCL1 (*Figure 7—figure supplement 1C*). In wild-type HeLa cells, Bim associated primarily with Mcl-1. Introduction of αMCL1 resulted in displacement of Bim from Mcl-1, with modest compensatory sequestration of Bim by Bcl-2. In Bcl-2+ cells, Bim is redistributed and preferentially binds Bcl-2 rather than Mcl-1, likely due to the stoichiometric excess of Bcl-2, and αMCL1 has no effect. The cell-killing activity of αMCL1 with αBCL2 in wild-type, Mcl-1+ and Bcl-2+ cells is consistent with these data; inhibition of both Mcl-1 and Bcl-2 in these settings likely overwhelms BOP sequestration, and a higher proportion of Bim and other activator BOPs may be free to interact with Bak and Bax, inducing apoptosis.

## Designed inhibitors elucidate the dependence of human cancer cell lines on pro-survival BCL2 homologs

Next, we set out to define functional BCL2 dependency profiles of other cancer cell lines using a larger set of our designed inhibitors. Apoptotic resistance in melanoma is thought to act via Bfl-1 (*Hind et al., 2015*), and likewise in glioblastoma via Bcl-2 (*Weller et al., 1995*) and Bcl-xL (*Nagane et al., 2000*). Further, oncogenic EGFR mutations in glioblastoma are associated with apoptotic resistance via increased Bcl-xL expression (*Latha et al., 2013*). Therefore, melanoma and EGFR-modified glioblastoma cell lines provide diverse contexts to test the BCL2-profiling capacity of the designed proteins.

In all cell lines, single inhibitors again were unable to induce apoptosis. While SK-MEL-5 were overall more resistant to apoptosis, LOX-IMVI melanoma cells were sensitive to double combinations that included αMCL1 and triple combinations (*Figure 8A*). αBFL1 with αBCL2 or αBCLXL had less effect, indicating that Mcl-1 plays a more critical role in survival than Bfl-1 in LOX-IMVI, in contrast to mRNA profiling suggesting the opposite (*Hind et al., 2015*). All glioblastoma cell lines showed similar trends in response to all combinations, while EGFR variants were in some instances more resistant than parental (*Figure 8B*). Sensitivity to many different double combinations suggests that in these contexts, pro-survival homologs may have more redundant biological function and resist apoptosis via 'mode 1' interactions with the pan- or partially-specific BOPs (*Llambi et al., 2011*).

To more fully assess the capacity of the designed inhibitors to determine BCL2 profiles, we tested them alongside existing, selective BH3-mimetics in a larger number of cell lines from one type of cancer. In previous studies, colon cancers showed a variable response to small-molecule-mediated Bcl-xL inhibition, and RNAi experiments identified Mcl-1 as a resistance factor (*Zhang et al., 2015*). To determine whether the Mcl-1 antagonism could render colon cancers sensitive to Bcl-xL neutralization and assess the influence of other pro-survival homologs on survival, we modified a panel of seven colon cancer lines to inducibly express either αMCL1 or αBFL1, and treated them with small molecules to selectively inhibit Bcl-2 (ABT-199), Bcl-xL (A-1331852), or Bcl-2 and Bcl-xL simultaneously (ABT-263).

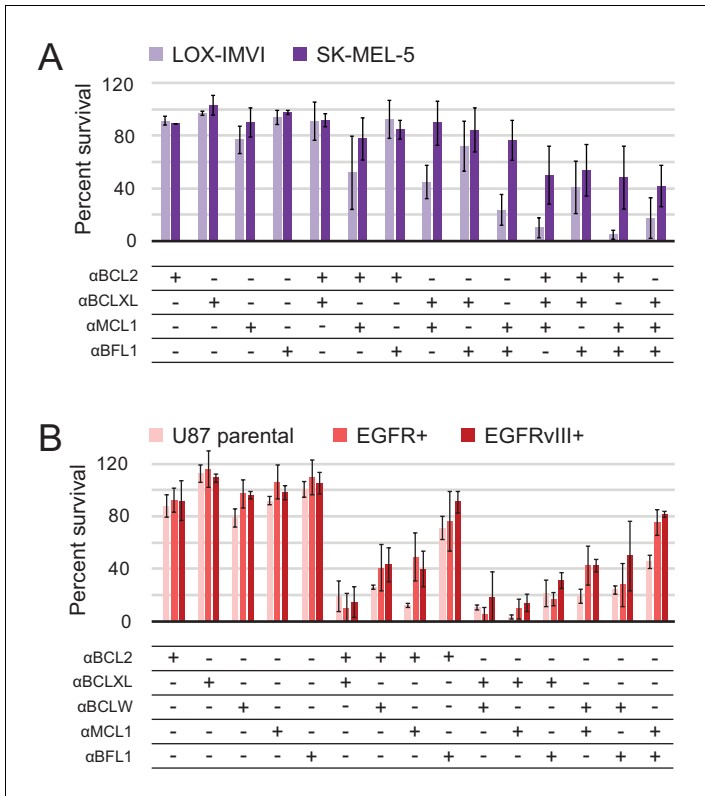

**Figure 8.** Determination of functional BCL2 profiles in melanoma and glioblastoma cell lines. (**A**) Melanoma and (**B**) glioblastoma cell lines were transduced with constructs for designed inhibitor expression and viability was assayed after 72 hr (mean ± SD; for melanoma, n = 2 to 4; for glioblastoma, n = 4). See also *Figure 9—figure supplement 1D* for Western blot analysis of pro-survival proteins.

The following source data is available for figure 8:

**Source data 1.** Source data relating to *Figure 8*.

Inhibiting a single pro-survival homolog had little effect on short-term survival; only SW48 cells showed greater than a 50% decrease in viability after treatment with A-1331852, consistent with a previous study showing SW48 is sensitive to Bcl-xL inhibition (*Zhang et al., 2015*; *Figure 9A*). Combined inhibition of both Mcl-1 and Bcl-xL caused nearly complete cell death after 24 hr in all colon cancers except HCT-116; further analyses showed that αMCL1-mediated Mcl-1 inhibition strongly sensitizes most colon cancers to A-1331852 (and to a lesser extent ABT-263), with a 4.6-fold or greater decrease in $EC_{50}$ values observed in all cell lines except HCT-116 (*Figure 9—figure supplement 1A–B*). All other combinations had much smaller effects. Thus, in contrast to gliobastoma where pro-survival proteins appeared largely redundant, inhibition of two pro-survival proteins was required and sufficient for cell killing. These results suggest that in context of colon cancer, pro-survival proteins may resist apoptosis primarily via 'mode 2' inhibition of the direct effector Bak, which interacts preferentially with Mcl-1 and Bcl-xL (*Llambi et al., 2011*). As αMCL1 targets Mcl-1 in a manner more akin to a drug (i.e. antagonism) compared to RNAi, our data provide further evidence that treatment strategies involving Mcl-1 and Bcl-xL inhibition could be effective in these malignancies.

In long-term survival assays, αMCL1 had negligible effect, but remarkably, αBFL1 caused a significant decrease in RKO cell survival (63 ± 4% decrease; *Figure 9B*). Thus, long-term assays detect sensitivities that short-term assays miss, on a timescale that may provide a more informative preview of therapy. Overall, these data show the utility and sensitivity of our inhibitors in establishing the critical survival factors in colon cancer.

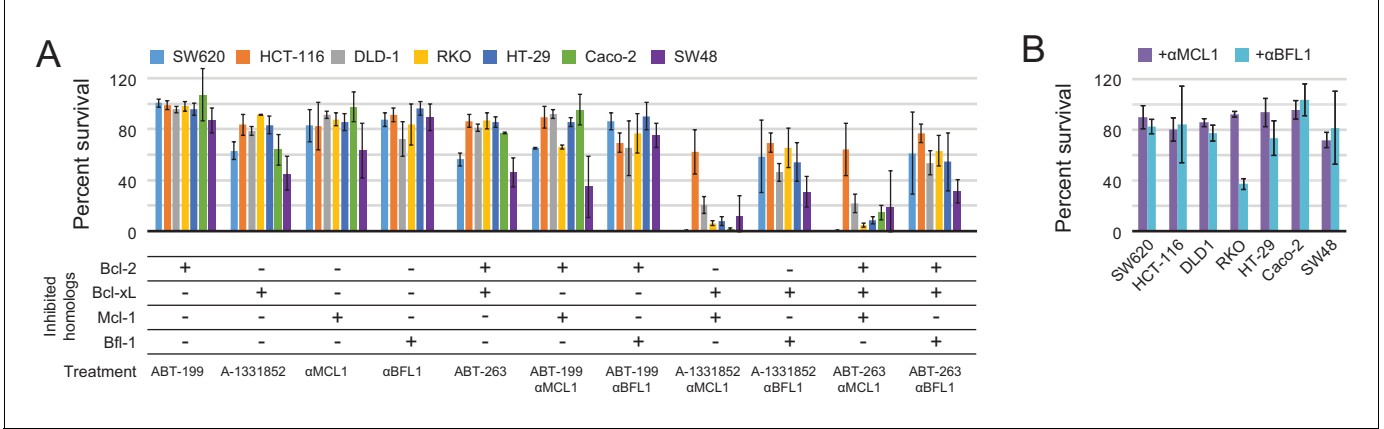

**Figure 9.** Determination of functional BCL2 profiles in colon cancer cell lines. (**A**) Colon cancers were treated with small molecule drugs (2 µM) and/or doxycycline to induce expression of designed inhibitors, as indicated, and viability was assayed after 24 hr (mean ± SD; n = 3). (**B**) Long-term survival was assessed after seven to ten days of doxycycline-induced expression of αMCL1 or αBFL1 (mean ± SD; n = 3).

The following source data and figure supplement are available for figure 9:

**Source data 1.** Source data relating to *Figure 9* and *Figure 9—figure supplement 1*.

**Figure supplement 1.** Drug titrations and long-term survival assays in colon cancers.

## Discussion

This work offers the first complete set of specific inhibitors for each of the six pro-survival BCL2 proteins, including the first reported specific inhibitors for Bcl-w and Bcl-B. Our designed inhibitors exhibit greater specificity and in many cases higher affinity than small molecule alternatives, and have advantages unique to their protein composition. For example, the designed proteins can be easily modified for added functionality, such as adding a mitochondrial targeting sequence, or fusing an E3 ligase to each design to catalyze degradation of their target BCL2 proteins. The designed protein inhibitors can be genetically encoded, enabling spatial and temporal control of expression, and have distinct advantages over broadly eliminating the target BCL2 protein using CRISPR- or RNAi-mediated knockdown or knockout. The designs can be used to probe mechanism; we show that specific inhibitors cause the redistribution of a representative BOP, Bim, and the approach can be used to probe other BOPs and compare 'mode 1' versus 'mode 2' inhibition of apoptosis (*Llambi et al., 2011*). Some BCL2 proteins translocate from the cytosol to the mitochondrial membrane in response to apoptotic stimuli, and the effect of inhibition in these different compartments can be probed by localizing the designed inhibitors with the appropriate targeting sequences and inducing expression before and after apoptotic stimuli. The designed proteins can also be used to distinguish interactions at sites other than the BH3-binding groove; for example, Bcl-xL is thought to interact with p53 at a site opposite the BH3-binding groove (*Petros et al., 2004*), and Bcl-2 is reported to interact with the IP3 receptor in the endoplasmic reticulum via Bcl-2's BH4-domain (*Rong et al., 2009*). These studies are simply impossible with CRISPR or RNAi strategies.

Our computational design calculations using the stable de novo designed protein BINDI as a starting point enabled us to achieve, in the cases of Mcl-1 and Bcl-2, high specificity and affinity immediately following design, and in the cases of Bcl-xL, Bcl-w, Bfl-1 and Bcl-B, superior starting points for optimization compared to a single, pan-specific construct. Our success in designing not one but six specific inhibitors demonstrates the generality of the design method. We are not aware of any precedent among designed proteins or indeed in nature for two sets of six closely related proteins in which each protein in one set has the extremely high specificity (100–100,000 fold) for a unique member of the other set.

As confirmed by biochemical analyses and X-ray crystal structures, the designed proteins engage the BH3-binding grooves of their specific target pro-survival BCL2 family members. The designs

were used to determine the BCL2-dependence of different cancers, providing a more direct guide for therapy than knockdown/knockout strategies or mRNA analysis by mimicking the mechanism of action of BCL2-targeting small molecule drugs. While mRNA profiling suggests that Bfl-1 confers apoptotic resistance in SK-MEL-5 and LOX-IMVI melanomas (*Hind et al., 2015*), our combinatorial antagonism of pro-survival homologs indicates that Mcl-1 plays a more critical role and further discriminates between sensitive LOX-IMVI and resistant SK-MEL-5. We also provide further evidence that many colon cancers are dependent on Mcl-1 and Bcl-xL for survival; mRNA profiling indicates Mcl-1 and Bcl-xL are indeed more prevalent than other BCL2 homologs in many colon cancers, but resistant HCT-116 is indistinguishable from sensitive lines like Caco-2 and HT-29 (*Placzek et al., 2010*). Further, the detection of RKO sensitivity to Bfl-1 inhibition highlights the capacity of the designed inhibitors to illuminate unique BCL2 profiles, even among cancers with similar general characteristics.

More generally, computationally designed inhibitors enable the investigation of the biological roles of specific protein interactions with the high spatio-temporal control that can be achieved with tissue-specific and inducible promoters. Competing approaches offer less control. The distribution of small molecules is difficult to spatially or temporally control in vivo, and broadly eliminating the protein of interest with CRISPR or RNAi cannot probe interactions with a specific interface or capture mechanistic intricacies. This work demonstrates that high affinity and specificity protein inhibitors can be designed for each member of a closely-knit protein family, providing a unique opportunity to probe the importance of individual protein interactions.

## Materials and methods

### Protein design and purification

Proteins were designed using the ROSETTA software suite, and genes for designed proteins and target Bcl-2 homologs were synthesized by oligo assembly or by commercial suppliers. All proteins were expressed in *E. coli* and purified via metal affinity chromatography followed by gel filtration. BCL2 homologs were enzymatically biotinylated in vitro with BirA. Purified designed proteins were screened for binding to BCL2 homologs with bio-layer interferometry.

### Protein optimization

Designed proteins were optimized by yeast surface display. Gene sequences were diversified by overlapping PCR for SSM libraries (*Procko et al., 2013*), oligo assembly with degenerate primers for combinatorial libraries, or by error-prone PCR. Gene libraries were expressed in yeast for surface display and sorted for binding to labeled target homolog in the presence of unlabeled competitors (*Chao et al., 2006*). Deep sequencing analysis of sorted populations (using adapted scripts from Enrich; *Fowler et al., 2011*) informed manual optimization and combinatorial library design.

### Cell line generation, authentication and mycoplasma testing

Mouse embryonic fibroblasts were generated from E13-E14.5 embryos derived from *CreERT2/Bcl-x^{fl/fl}/Mcl-1^{fl/fl}* C57BL/6 mice (*Kelly et al., 2014*) and immortalized (at passage 2–4) with SV40 large T antigen. HeLa cells (originally obtained from ATCC, RRID:CVCL_0030) were generously provided by Dusty Miller at the Fred Hutchinson Cancer Research Center (Seattle, WA). Melanoma cell lines (LOX-IMVI, RRID:CVCL_1381; SK-MEL-5, RRID:CVCL_0527) were purchased from the National Cancer Institute (NCI). Glioblastoma cells lines were generously provided by Paul Mischel at the Ludwig Institute for Cancer Research (San Diego, CA); U87 (originally obtained from ATCC; RRID:CVCL_0022) were modified to express EGFR and variant EGFRvIII as described by *Wang et al. (2006)*. SW620 (originally obtained from ATCC, RRID:CVCL_0547), HCT-116 (ATCC, RRID:CVCL_0291), DLD1 (ATCC, RRID:CVCL_0248), RKO (ATCC, RRID:CVCL_0504), HT-29 (ATCC, RRID:CVCL_0320), Caco-2 (ATCC, RRID:CVCL_0025), and SW48 (ATCC, RRID:CVCL_1724) colon cancer cell lines were generously provided by John Mariadason at the Olivia Newton-John Cancer Research Institute.

For colon cancer cell lines, authentication was performed using the Promega StemElite ID System (Promega, Madison, WI) at the Queensland Institute of Medical Research (QMIR, Queensland, Australia) DNA Sequencing and Fragment Analysis Facility (January 2013). All colon cancer cell lines and parent MEF cell lines tested negative for mycoplasma by the MycoAlert Mycoplasma Detection Kit

(Lonza). HeLa, melanoma and glioblastoma cell lines have not been authenticated in our hands, and each tested negative for mycoplasma by the MycoFluor Mycoplasma Detection Kit (Thermo Fisher Scientific, Waltham, MA).

MEF and HeLa cells were retrovirally infected with constructs for constitutive expression of BCL2 pro-survival homologs and selected with FACS (MEF) or geneticin (HeLa). In MEFs, endogenous Mcl-1 and Bcl-xL were deleted via Cre-Lox recombination (*Kelly et al., 2014*). Engineered MEF and HeLa cells, colon cancer, glioblastoma and melanoma cells were lentivirally infected with constructs for constitutive or inducible expression of designed inhibitors (*Aubrey et al., 2015*). Infected cells were selected with antibiotics or FACS, and stable cell lines were cultured.

### Survival assays

For short-term survival assays, engineered MEFs and colon cancer cells were treated with doxycycline to induce designed protein expression and/or small molecule drugs at indicated final concentrations. Viability was assayed after 24 hr. Engineered HeLa, melanoma and glioblastoma cells were transiently transduced with designed inhibitors. Viability was assayed after 72 hr.

To assay long-term survival, MEF and colon cancers were sparsely plated, then treated with doxycycline to induce designed protein expression the next day and approximately every 48 hr for the next seven to ten days. Media was aspirated and colonies were stained and manually counted.

### Additional methods

Please see the Appendix I for a more detailed description of methods.

### Accession numbers

The crystal structure factors and coordinates of αMCL1•Mcl-1 (PDB 5JSB) and αBCL2•Bcl-2 (PDB 5JSN) have been deposited in the Protein Data Bank. Deep sequencing data, both raw and processed files, have been deposited in the National Center for Biotechnology Information Gene Expression Omnibus repository with accession number GSE80194.

## Acknowledgements

Many thanks to Andreas Strasser and Gemma Kelly who provided mice from which MEFs were harvested, and John Mariadason who provided colon cancer cell lines and guidance. This work was supported by the NIH (P41GM103533, R01 GM115545, R01 CA158921-04), DTRA (HDTRA1-10–0040), and HHMI (HHMI-027779). SAB is supported by the NSF GRFP. DAS is a PEW Latin-American fellow in the biomedical sciences and a CONACyT postdoctoral fellow. WDF and EFL were supported by grants from Worldwide Cancer Research (15–0025) and Cancer Council of Victoria (1057949). EFL was supported by a Career Development Fellowship from the National Health and Medical Research Council of Australia (1024620) and a Future Fellowship from the Australian Research Council (FT150100212). The Olivia Newton-John Cancer Research Institute acknowledges the Operational Infrastructure Support Program of the Victorian Government, Australia for partial funding of this project.

## Additional information

### Funding

| Funder | Grant reference number | Author |
| --- | --- | --- |
| National Institutes of Health | P41GM103533 | Stephanie Berger<br>Erik Procko<br>David Baker |
| Defense Threat Reduction Agency | HDTRA1-10-0040 | Stephanie Berger<br>Erik Procko<br>David Baker |
| Howard Hughes Medical Institute | HHMI-027779 | Stephanie Berger<br>Erik Procko<br>David Baker |

| National Science Foundation | Graduate Research Fellowship Program | Stephanie Berger |
| Worldwide Cancer Research | 15-0025 | Erinna F Lee<br>W Douglas Fairlie |
| Cancer Council Victoria | 1057949 | Erinna F Lee<br>W Douglas Fairlie |
| Pew Charitable Trusts | | Daniel-Adriano Silva |
| Consejo Nacional de Ciencia y Tecnología | | Daniel-Adriano Silva |
| National Health and Medical Research Council | 1024620 | Erinna F Lee |
| Australian Research Council | FT150100212 | Erinna F Lee |
| National Institutes of Health | R01 GM115545 | Betty W Shen<br>Barry L Stoddard |
| National Institutes of Health | R01 CA158921-04 | Daciana Margineantu<br>David M Hockenbery |
| Victorian Government, Australia | Operational Infrastructure Support Program | Erinna F Lee<br>W Douglas Fairlie |

The funders had no role in study design, data collection and interpretation, or the decision to submit the work for publication.

## Author contributions

SB, EP, DM, EFL, BWS, WDF, Conception and design, Acquisition of data, Analysis and interpretation of data, Drafting or revising the article; AZ, Acquisition of data, Analysis and interpretation of data, Drafting or revising the article; D-AS, Conception and design, Drafting or revising the article, Contributed unpublished essential data or reagents; KC, Performed related immunoblotting, Acquisition of data; MJH, Generated constructs for inducible designed protein expression, Contributed unpublished essential data or reagents; J-MG, Synthesized A-1331852, Contributed unpublished essential data or reagents; RJ, Performed MS experiments, Conception and design; MJM, Generated constructs for inducible designed protein expression, Conception and design; GL, Conception and design, Drafting or revising the article; TND, PSS, Supervised research, Conception and design; BLS, Conception and design, Analysis and interpretation of data; DMH, DB, Conception and design, Analysis and interpretation of data, Drafting or revising the article

## Author ORCIDs

Stephanie Berger, http://orcid.org/0000-0002-3738-5907
Guillaume Lessene, http://orcid.org/0000-0002-1193-8147
Trisha N Davis, http://orcid.org/0000-0003-4797-3152
David Baker, http://orcid.org/0000-0001-7896-6217

## Additional files

### Supplementary files

• Supplementary file 1. Data tables. (A) Summary of computational designs selected for protein production and biochemical analysis. (B) Sequences of computational designs and optimized variants. (C) Crystallographic data collection and refinement statistics. (D) Protein cross-linking of the αMCL1-Mcl-1 complex. (E) Sort conditions for all in vitro evolution experiments. (F) Mutation summary for evolved variants.

• Supplementary file 2. CDP design models. PDB models of all computationally designed proteins (CDPs). Please see *Supplementary file 1A* for descriptions and computational statistics.

### Major datasets

The following dataset was generated:

| Author(s) | Year | Dataset title | Dataset URL | Database, license, and accessibility information |
|---|---|---|---|---|
| Berger S, Procko E, Baker D | 2016 | Computationally designed, high specificity inhibitors delineate the roles of BCL2 family proteins in cancer | https://www.ncbi.nlm.nih.gov/geo/query/acc.cgi?acc=GSE80194 | Publicly available at the NCBI Gene Expression Omnibus (accession no: GSE80194). |

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

## Appendix 1

## Additional methods

### Computational methods: General

ROSETTA software can be downloaded from www.rosettacommons.org and is available free to academic users. Online documentation can be found at: http://www.rosettacommons.org/manuals/archive/rosetta3.5_user_guide/index.html

and instructions for RosettaScripts syntax is available at: http://www.rosettacommons.org/docs/latest/scripting_documentation/RosettaScripts/RosettaScripts

A comprehensive list of command line options for ROSETTA can be found at: www.rosettacommons.org/docs/latest/full-options-list

### RosettaScripts framework

All computational protocols were executed from within the RosettaScripts framework, which enables the user to piece together select portions of ROSETTA code in order to generate project-specific protocols (*Leaver-Fay et al., 2011*; *Fleishman et al., 2011a*). An example of a command line executed to launch ROSETTA employing a RosettaScripts protocol is as follows:

```
/path/rosetta_scripts.default.linuxgccrelease
-database /path/main/database
-parser:protocol rosetta_scripts_protocol.xml
-in:file:native BINDI.pdb
-nstruct 3
-ex1
-ex2
-ignore_zero_occupancy false
```

An example of a RosettaScripts XML protocol is found below, under 'Computational Methods: Design with ROSETTA.'

## Computational methods: Generating docked configurations of BINDI in the hydrophobic groove of BCL2 homologs

### Input models

The following crystallographic models of ligand-bound BCL2 homologs, found in the Protein Data Bank, were used to manually graft side chains onto a fixed backbone, as described below: 2PQK (Mcl-1•Bim-BH3), 3PK1 (Mcl-1•Bax-BH3), 3KZ0 (Mcl-1•MB7 peptide), 2XA0 (Bcl-2•Bax-BH3), 4AQ3 (Bcl-2•phenylacylsulfonamide), 4IEH (Bcl-2•sulfonamide), 4LVT (Bcl-2•Navitoclax), 1PQ1 (Bcl-xL•Bim-BH3), 2YQ6 (Bcl-xL•BimSAHB), 2YQ7 (Bcl-xL•BimLOCK), 3PL7 (Bcl-xL•Bax-BH3), 4BPK (Bcl-xL•α/β-Puma-BH3), 4K5A (Bcl-w•DARPin) 3I1H (Bfl-1•Bak-BH3), and 4B4S (Bcl-B•Bim-BH3).

Additional models of Bcl-w were generated for input into an automated motif grafting protocol described below. The Bcl-w sequence was threaded onto structurally analogous positions in existing crystallographic models of other BCL2 homologs. Only models bound to helical motifs were used: 1PQ1, 2BZW (Bcl-xL•Bad-BH3), 2YJ1 (Bcl-xL•α/β-Puma-BH3), 2YQ6, 2YQ7, 3FDL (Bcl-xL•Bim-BH3), 4A1U (Bcl-xL•designed α/β-foldamer), 4A1W (Bcl-

xL•designed α/β-foldamer), 4BPK, 4HNJ (Bcl-xL•Puma-BH3), and 4OYD (BHRF1•BINDI). The TM-align software (*Zhang and Skolnick, 2005*) was used to generate structural alignments. Each new Bcl-w model then underwent constrained backbone and side chain minimization in the presence of the bound helical motif borrowed from the initial crystallographic model. The Bcl-w•helix complex was then aligned to a common 20-amino-acid truncated BH3-motif (truncatedBH3.pdb; KEKYIAAMLRAIGDIFNAIM) using PyMOL (Schrödinger). New PDB files of each Bcl-w model positioned to bind the common BH3-motif were saved and input as 'context' in the automated motif grafting protocol described below.

Additional conformations of the partially-nonspecific Mcl-1-targeting binder, M-CDP02, were sampled by submitting the M-CDP02 sequence to ROSETTA's ab initio structure prediction protocol (*Rohl et al., 2004*). Of 30,200 generated models, any having greater than 2.5 Å RMSD relative to the starting model of M-CDP02 were discarded. 250 models with the most favorable (lowest) total score in ROSETTA energy units were input as 'scaffolds' for the automated motif grafting protocol described below.

## Manual side-chain grafting on a fixed backbone

A suitable helical region of the BINDI protein (PDB 4OYD chain B) was aligned to the BH3-motif ligand in crystallographic models of each BCL2 pro-survival homolog, using PyMOL (Schrödinger; PDB IDs noted in *Supplementary file 1A*). If the target structure was bound to an unnatural ligand, such as a small molecule or α/β-foldamer, the model of the pro-survival homolog was first aligned to an alternative structure bound to a helical BH3 motif, which then served as a guide for structural alignment of BINDI. The structural alignment was visually inspected, and any docked configurations with backbone clashes between the scaffold protein and BCL2 homolog were discarded. Side chain clashes were tolerated, as they may be resolved later by sequence design of the scaffold and by rotamer repacking on the target. Important interfacial residues from each BH3-motif were transferred, or grafted, to the aligned BINDI scaffold and kept fixed during the subsequent design protocol; these 'hotspot' residues per model are listed in *Supplementary file 1A*. A new PDB file containing the partially mutated scaffold bound to the target homolog was saved and used as the input for ROSETTA-based design.

## Computational motif grafting on a fixed backbone

Grafting is a 'seeded interface' protein design approach (*Correia et al., 2010*), in which a small motif of known structure that binds to a target site of interest is used to initiate the protein design process. The motif is then grafted (i.e. embedded) into a larger protein scaffold, which both stabilizes the structure of the small motif and contributes additional favorable interactions with the target protein. We have implemented a new computational grafting protocol as the MotifGraft mover in RosettaScripts, described in detail by *Silva et al. (2016)*. The input of MotifGraft is composed of three structures: (1) the motif, which is a protein fragment that is intended for grafting in a new protein scaffold; (2) the context, which is the macromolecule interacting with the motif; and (3) the target scaffolds, which are protein scaffolds that the protocol will use to search insertion points for the motif. The goal of MotifGraft is to find fragments in the target scaffolds that are geometrically compatible with the specified motif(s), and then replace those fragments with the motif(s) itself. In this case, the parameters of grafting were settled to perform full backbone alignment of the input motif, with a maximum RMSD of the backbone of 3.0 Å and RMSD for the endpoints of 2.0 Å. For the input motif 'truncatedBH3.pdb' the hotspot residues were defined as: LEU-9, ILE-12, GLY-13, ASP-14, PHE-16 and ASN-17. The protocol was instructed to revert all other residues to their native identities in the target scaffold. No clashes between the grafted design and the context protein were allowed. The following mover was added to the XML script to implement this protocol within the RosettaScripts framework:

```
<MotifGraft name="motif_grafting"
```

```
        context_structure="%%context%%"
        motif_structure="truncatedBH3.pdb"
        RMSD_tolerance="3.0"
        NC_points_RMSD_tolerance="2.0"
        clash_score_cutoff="0"
        clash_test_residue="ALA"
        hotspots="9:12:13:14:16:17"
        combinatory_fragment_size_delta="0:0"
        max_fragment_replacement_size_delta="0:0"
        full_motif_bb_alignment="1"
        allow_independent_alignment_per_fragment="0"
        graft_only_hotspots_by_replacement="0"
        only_allow_if_N_point_match_aa_identity="0"
        only_allow_if_C_point_match_aa_identity="0"
        revert_graft_to_native_sequence="1"
        allow_repeat_same_graft_output="1"/>
```

## Computational methods: Design with ROSETTA

An example RosettaScripts XML file used for computational design after manual or computational motif-grafting is below. The script is annotated with brief descriptions, and the indicated pieces of ROSETTA code were implemented in the order listed in the '<PROTOCOLS>' section below. Designs were generated and then filtered by the indicated metrics.

```
<dock_design>
<SCOREFXNS>
        <sfxn_std_cst weights=talaris2013>
            <Reweight scoretype=coordinate_constraint weight = 1.5/>
        </sfxn_std_cst>
</SCOREFXNS>
<TASKOPERATIONS>
        <InitializeFromCommandline name="init"/>
        <LimitAromaChi2 name="arochi2"/>
        <IncludeCurrent name="inclcur"/>
        <ExtraRotamersGeneric name="exrot" ex1="1" ex2="1" extrachi_cutoff="1"/>
# General task operations: don't mutate chain A (target homolog) or residues defined
as hotspots
        <OperateOnCertainResidues  name="restrict_chainA">  <ChainIs  chain=A/>
<RestrictToRepackingRLT/> </OperateOnCertainResidues>
        <OperateOnCertainResidues  name="rtr_hotspots">  <ResiduePDBInfoHasLabel
property="HOTSPOT"/> <PreventRepackingRLT/> </OperateOnCertainResidues>
        <RestrictToRepacking name=rtr/>
# Task operations related to core design: only select hydrophobic residues in core,
then try alternate hydrophobics but favor BINDI sequence.
        <JointSequence name="native" use_current = 0 use_native = 1 chain = 2 />
        <RestrictAbsentCanonicalAAS name="try_apolars" keep_aas="AFILMV"/>
        <SelectBySASA name=core mode="sc" state="monomer" probe_radius="2.0" cor-
e_asa = 0 surface_asa = 30 core = 1 boundary = 1 surface = 0 verbose = 1/>
        <RestrictIdentities name="design_apolars_only" identities="CYS,ASP,GLN,
GLU,GLY,HIS,LYS,ASN,PRO,ARG,SER,THR,TRP,TYR"/>
# Task operations related to interface design
        <DisallowIfNonnative name="dont_allow_PCWG" disallow_aas="CPWG"/>
        <SelectBySASA name=surface mode="sc" state="monomer" probe_radius="2.0"
core_asa = 0 surface_asa = 30 core = 0 boundary = 0 surface = 1 verbose = 1/>
# Task operations related to trying more hydrophilic residues at surface residues
currently having hydrophobic IDs
        <OperateOnCertainResidues  name="dont_design_polars">  <ResidueName3Isnt
name3=ALA,LEU,VAL,ILE,MET,PHE,TRP,GLY/>  <RestrictToRepackingRLT/>  </OperateOn-
CertainResidues>
        <SelectBySASA  name="only_scaffold_surface_and_non_interface"  mode="sc"
state="bound" probe_radius="2.0" core_asa = 0 surface_asa = 40 core = 0 boundary = 0
surface = 1 verbose = 1/>
```

```
            <RestrictAbsentCanonicalAAS name="try_polars" keep_aas="DEHKNQRSTY"/>
# Task operations related to trying mutation of serines with limited solvent accessi-
bility to small hydrophobics
            <RestrictAbsentCanonicalAAS name="try_small_hydrophobic" keep_aas=AV/>
            <OperateOnCertainResidues name="find_serines"> <ResidueName3Isnt name3=-
SER/> <RestrictToRepackingRLT/> </OperateOnCertainResidues>
            <SelectBySASA name=boundary mode="sc" state="bound" probe_radius="2.0"
core_asa = 0 surface_asa = 30 core = 1 boundary = 1 surface = 0 verbose = 1/>
</TASKOPERATIONS>
<MOVERS>
            <Prepack name=ppk scorefxn=talaris2013 jump_number = 0/>
            <PackRotamersMover  name=revert  scorefxn=talaris2013  task_operations="-
restrict_chainA,native,rtr_hotspots" />
            <FavorSequenceProfile  name=favor_native_for_core_design  scaling="prob"
weight = 1.5 use_native = 1/>
            <PackRotamersMover name=design_core scorefxn=talaris2013 task_operation-
s="init,inclcur,arochi2,exrot,core,restrict_chainA,try_apolars,design_apolar-
s_only,rtr_hotspots"/>
            <RepackMinimize  name=design_radius_8  repack_partner1 = 1  repack_part-
ner2 = 1 design_partner1 = 0 design_partner2 = 1 interface_cutoff_distance = 8.0 min-
imize_bb = 0 minimize_rb = 0 minimize_sc = 1 task_ope
rations="init,inclcur,arochi2,exrot,surface,dont_allow_PCWG,rtr_hotspots"/>
            <AddConstraintsToCurrentConformationMover      name=add_heavy_coor_epito-
pe_cst use_distance_cst = 0 coord_dev = 0.05 bound_width = 0.01 min_seq_sep = 8 max_-
distance = 12.0 cst_weight = 1.0 CA_only=
0 bb_only = 1/>
            <RepackMinimize name=design_radius_12 repack_partner1 = 1 repack_part-
ner2 = 1 design_partner1 = 0 design_partner2 = 1 interface_cutoff_distance = 12.0
minimize_bb = 0 minimize_rb = 1 minimize_sc = 1 task_op
erations="init,inclcur,arochi2,exrot,surface,dont_allow_PCWG,rtr_hotspots"/>
            <RepackMinimize  name= try_small_hphobic_at_serines design_partner2 = 1
design_partner1 = 0 minimize_bb = 0 minimize_rb = 1 minimize_sc = 1 interface_cu-
toff_distance = 1000 task_operations="init,in
clcur,arochi2,exrot,rtr_hotspots,find_serines,try_small_hydrophobic"/>
            <RepackMinimize   name=fix_surface_hydrophobics   design_partner2  =  1
design_partner1 = 0 minimize_bb = 0 minimize_rb = 1 minimize_sc = 1 interface_cu-
toff_distance = 1000 task_operations="init,incl
cur,arochi2,exrot,dont_design_polars,only_scaffold_surface_and_non_interface,
try_polars,rtr_hotspots"/>
            <RepackMinimize name=final_relax design_partner2 = 0 design_partner1 = 0
repack_partner1 = 1 repack_partner1 = 0 minimize_bb = 0 minimize_rb = 0 minimize_sc = 1
interface_cutoff_distance = 1000 tas
k_operations="rtr"/>
</MOVERS>
<FILTERS>
            <Ddg name=ddg scorefxn=talaris2013 threshold = 0 confidence = 1/>
            <BuriedUnsatHbonds name=unsat cutoff = 10 confidence = 1/>
            <ShapeComplementarity name=Sc min_sc = 0.45 confidence = 1/>
            <InterfaceHoles name="interfaceHoles" jump="1" threshold="200"/>
            <ScoreType  name="lr_elec"  scorefxn="talaris2013"  score_type="fa_elec"
threshold="1200"/>
            <ScoreType name="total_score" scorefxn="talaris2013" score_type="total_-
score" threshold="0" confidence="1"/>
            <Sasa name="sasa_general" threshold = 0 />
            <TotalSasa name="sasa_hydrophobic" threshold = 0 hydrophobic="1" report_-
per_residue_sasa="1"/>
</FILTERS>
<APPLY_TO_POSE>
</APPLY_TO_POSE>
<PROTOCOLS>
            <Add mover_name=ppk/>
            <Add mover_name=revert/>
            <Add mover_name=add_heavy_coor_epitope_cst/>
            <Add mover_name=design_radius_8/>
            <Add mover_name=design_radius_12/>
            <Add mover_name=try_small_hphobic_at_serines />
            <Add mover_name=fix_surface_hydrophobics/>
```

```
        <Add mover_name=favor_native_for_core_design/>
        <Add mover_name=design_core/>
        <Add mover_name=final_relax/>
        <Add filter_name=ddg/>
        <Add filter_name=Sc/>
        <Add filter_name=unsat/>
        <Add filter_name=total_score/>
        <Add filter_name=lr_elec/>
        <Add filter_name=sasa_general/>
        <Add filter_name=sasa_hydrophobic/>
</PROTOCOLS>
</dock_design>
```

## Computational Methods: Docking

First, optimized inhibitors were modeled using Rosetta by inputting the precursor CDP model, explicitly specifying the appropriate mutations, and relaxing the final complex using the FastRelax mover and allowing both backbone and side chain minimization. All non-cognate binding pairs were first generated manually in PyMol by aligning all complexes and simply creating new molecules comprising non-cognate pairs, then using Rosetta to relax each complex using the FastRelax mover and allowing both backbone and side chain minimization. The PatchDock protocol was used to generate approximate docked orientations, and the following RosettaScripts protocol was used for local and global docking:

```
<dock_design>
        <SCOREFXNS>
            <fullatom weights=beta symmetric = 0>
            </fullatom>
        </SCOREFXNS>
        <FILTERS>
            <Ddg name=ddg scorefxn=fullatom threshold = 0 jump = 1 repeats = 1
repack = 1 confidence = 1/>
            <Sasa name=sasa confidence = 0/>
            <ShapeComplementarity name=shape verbose = 1 confidence = 0 jump = 1/>
        </FILTERS>
        <MOVERS>
            <AtomTree name=docking_tree docking_ft = 1/> connect chains by their
geometric centres. Good for minimization
            <DockSetupMover name=setup_dock/>
            <DockingProtocol name=dock docking_score_high=fullatom low_res_proto-
col_only = 0 docking_local_refine = 0 dock_min = 1 ignore_default_docking_task = 0/>
        </MOVERS>
        <APPLY_TO_POSE>
        </APPLY_TO_POSE>
        <PROTOCOLS>
            <Add mover_name=docking_tree/>
            <Add mover_name=setup_dock/>
            <Add mover_name=dock/>
            <Add filter_name=ddg/>
            <Add filter_name=sasa/>
            <Add filter_name=shape/>
        </PROTOCOLS>
</dock_design>
```

## Protein purification

All commercially synthesized DNA constructs were codon-optimized for *E. coli* and purchased from Integrated DNA Technologies. Genes were assembled from oligo primers with Phusion

polymerase (New England Biolabs, Ipswich, MA) and cloned into pET29b (Novagen), adding a C-terminal 6-histidine tag. Protein was expressed in BL21*(DE3) *E. coli* and purified by metal affinity chromatography and size exclusion chromatography (SEC). Target Bcl-2 proteins with C-terminal avi-6His tags were similarly expressed and purified from *E. coli*, followed by enzymatic biotinylation using BirA (as per kit instructions from Avidity, Aurora, CO) and purification of biotinylated protein with metal affinity chromatography. All purified proteins were concentrated with ultrafiltration centrifugal devices (Sartorius, Goettingen, Germany), snap frozen in liquid nitrogen and stored at −80°C.

## Bio-layer interferometry

Data were collected on an Octet RED96 (ForteBio, Menlo Park, CA) and processed using the instrument's integrated software. All proteins were diluted from concentrated stock in binding buffer (10 mM HEPES [pH 7.4], 150 mM NaCl, 3 mM EDTA, 0.05% surfactant P20, 0.5% non-fat dry milk). Streptavidin-coated biosensors were dipped in wells containing biotinylated Bcl-2 proteins (25 nM) in binding buffer for 3–5 min for immobilization. After baseline measurement in buffer alone, binding kinetics were monitored by dipping the biosensors in wells containing defined concentrations of the designed protein (association), then dipping sensors back into baseline wells (dissociation). Titrations were done in triplicate and kinetic constants were determined from the mathematical fit of a 1:1 binding model.

## Protein optimization

SSM libraries were generated with overlap PCR (*Procko et al., 2013*), using Phusion polymerase and custom degenerate primers to introduce mutations to NNK at each codon. Mutations with highest enrichment in the sorted SSM (fitness for high affinity, specific binding to the target homolog) were combined in combinatorial libraries, generated by oligo assembly with primers having degenerate codons. The diversity of all combinatorial libraries was limited to less than $10^7$ variants. GeneMorph II Random Mutagenesis kit (Agilent Technologies, Santa Clara, CA) was used to introduce up to three random mutations in F-ECM04 and B-ECM01 genes. DNA libraries comprising genes for the initial designed protein sequence and related variants were cloned into the pETCON plasmid (*Fleishman et al., 2011b*), transformed into yeast, and expressed as fusions with N-terminal Aga2p for surface display and a C-terminal myc-tag in the EBY100 strain (*Chao et al., 2006*). Yeast libraries were grown in minimal media selective for the yeast strain (-ura) and the transforming plasmid (-trp), and protein expression was induced with 2% galactose. Surface expression was detected with anti-myc-FITC (Immunology Consultants Laboratory, Portland, OR), and binding to biotinylated Bcl-2 proteins after co-incubation for 0.5–2 hr at 22°C was detected with phycoerythrin-streptavidin (Invitrogen, Calsbad, CA). Yeast were sorted with an Influx (BD Biosciences, San Jose, CA) or SH800 (Sony Biotechnology, Inc., San Jose, CA) cell sorter and either plated on solid media for isolating and sequencing individual clones (Bcl-w-targeting design screen, combinatorial and epPCR libraries), or pelleted for batch DNA extraction and deep sequencing (SSM libraries).

## Deep sequencing analysis

Yeast were lysed with 125 U/ml Zymolase at 37°C for 5 hr, and DNA was harvested (Zymoprep kit from Zymo Research). Genomic DNA was digested with 2 U/µl Exonuclease I and 0.25 U/µl Lambda exonuclease (New England Biolabs) for 90 min at 30°C, and plasmid DNA purified with a QIAquick kit (Qiagen, Hilden, Germany). DNA was deep sequenced with a MiSeq sequencer (Illumina, San Diego, CA): genes were PCR amplified using primers that

annealed to external regions within the plasmid, followed by a second round of PCR to add flanking sequences for annealing to the Illumina flow cell oligonucleotides and a 6 bp sample identification sequence. PCR rounds were 12 cycles each with high-fidelity Phusion polymerase. Barcodes were read on a MiSeq sequencer using either a 300-cycle or 600-cycle reagent kit (Illumina), and sequences were analyzed with adapted scripts from Enrich (*Fowler et al., 2011*).

## Circular dichroism

CD spectra were recorded with a J-1500 Circular Dichroism Spectrometer (JASCO, Easton, MD). Proteins were at 10 µM in DPBS free of $MgCl_2$ and NaCl (Life Technologies, Calsbad, CA), and data were collected at 25°C.

## Crystal structure determination and refinement

Designs with C-terminal 6-His tags and untagged pro-survival homologs were expressed independently, and lysates of cognate pairs were co-purified with NiNTA affinity chromatography and SEC. Initial crystallization trials employed commercial screens using a MOSQUITO robot. Two conditions from Wizard I and II screen (indexes F4 and F6) yielded small crystals from the αMCL1•Mcl-1 complex that were reproducible in the mosquito tray but were not transferable to larger 24-well trays. Precipitant and pH optimizations using MOSQUITO trays yielded diffracting crystals in buffer conditions 1–1.3 M sodium citrate, 100 mM CHES, pH 9.5, which were then cryoprotected with paratone oil prior to flash freezing. 30% Jeffamine ED-2001, 100 mM HEPES, pH 7.0 (index D3) yielded crystals for αBCL2•Bcl-2, which were looped directly from the mother liquid and flash froze in liquid nitrogen. Data were collected either using an in house Rigaku MicroMax-007HF rotating anode generator equipped with a Saturn CCD detector or from beam line BL 5.0.2 at the Advance Light Source synchrotron facility at the Laurence Berkeley National laboratories. Datasets were integrated and scaled using HKL2000 (*Otwinowski and Minor, 1997*). Structures were determined using molecular replacement (PHASER, *McCoy et al., 2007*), RRID:SCR_014219) with 4LVTA and 2XA0A (Bcl-2), 3KZOA (Mcl-1) and computational models of αMCL1 and αBCL2. Refinements (by REFMAC5, (*Skubák et al., 2004*), RRID:SCR_014225) were conducted using the CCP4 program suite (*Winn et al., 2011*), RRID:SCR_007255) with CCP4i interface (*Potterton et al., 2003*). Model-building was carried out with COOT (*Emsley and Cowtan, 2004*), RRID:SCR_014222).

## Protein cross-linking and mass-spectrometric analysis

17 µg 3KZO-Y49 plus 22 µg Mcl1 were mixed in HB150 buffer (40 mM HEPES, 150 mM NaCl, 1 mM DTT, pH 7.5) in a final volume of 90.5 µL. Cross-linker concentration was brought to 0.86 mM by adding 14.5 mM DSS (disuccinimidyl suberate), DSG (disuccinimidyl glutarate) or BS3 bis(sulfosuccinimidyl)suberate. The reaction was allowed to proceed for 2 min at 25°C before quenching with 10 µL 500 mM $NH_4HCO_3$. Cross-linked proteins were reduced with 10 mM dithiothreitol (DTT) at 37°C for 30 min followed by 30 min alkylation at room temperature with 15 mM iodoacetamide (IAA). 25% vol of distilled water was added to the reactions prior to digestion with trypsin at a substrate-to-enzyme ratio of 60:1 overnight at room temperature with shaking. Digested samples were acidified with 5M HCl prior to being stored at −80°C until analysis. MS analysis was performed on a Q-Exactive (Thermo Fisher Scientific) and analyzed using the Kojak (version 1.4.2) cross-link identification software as previously described (*Zelter et al., 2015*; *Hoopmann et al., 2015*).

## MEF-derivative cell line generation

Retroviral expression constructs in the pMIG vector (Murine Stem Cell Virus-IRES-GFP) expressing each FLAG-tagged pro-survival protein were transiently transfected using Lipofectamine (Invitrogen), into Phoenix ecotropic packaging cells. Filtered virus-containing supernatants were used to infect the MEFs by spin inoculation as previously described (Lee et al., 2008). Cells stably expressing each pro-survival protein were selected by sorting GFP$^{+ve}$ cells 24 hr after spin inoculation and protein expression verified by Western blotting using an anti-FLAG antibody (Sigma-Aldrich, St. Louis, MO; RRID:AB_439687). Following verification of exogenous pro-survival protein expression, each cell line was treated with 1 µM Tamoxifen (Sigma-Aldrich) to enable deletion of endogenous Mcl-1 and Bcl-xL. Deletion of endogenous Mcl-1 and Bcl-xL was shown by Western blotting using anti-Mcl-1 (Rockland Clone, Limerick, PA; RRID:AB_2266446) and anti-Bcl-xL (BD Transduction Laboratories, RRID:AB_398070) antibodies. All Western blots were probed with anti-actin antibody (Sigma-Aldrich, RRID:AB_476697) to verify uniform loading. Cells were maintained in DME Kelso medium supplemented with 10% (v/v) fetal bovine serum, 250 mM L-asparagine and 50 mM 2- mercaptoethanol.

## HeLa-derivative cell line generation

HeLa cells were transfected with pSFFV vectors encoding human Mcl-1, Bcl-2, Bcl-xL, or empty vector (Neo) and selected with 1 mg/ml geneticin for 48 hr. Cells were maintained afterwards in DMEM with 10% (v/v) fetal bovine serum (FBS) supplemented with 500 µg/ml geneticin. Increased expression of pro-survival BCL2 proteins was confirmed by Western blotting using anti-Bcl-2 (Santa Cruz Biotechnology, Dallas, TX; RRID:AB_626736), anti-Bcl-xL (Santa Cruz Biotechnology, RRID:AB_630917), and anti-Mcl-1 (GeneTex, Irvine, CA; RRID: AB_377762) antibodies.

## Lentiviral infection

Inducible αMCL1 and αBFL1 constructs were generated in a lentiviral vector described in Aubrey et al. (2015). Ligand expression is linked via the T2A peptide to mCherry fluorescent reporter protein. Lentiviral particles were produced by transient transfection of 293T cells (AATC, RRID:CVCL_0063) with plasmid DNA along with the packaging constructs pMDL, pRSV-rev and pVSV-G using calcium chloride precipitation. Viral supernatants were then filtered prior to target cell transduction. For infection of MEFs and colon cancer cell lines, equal volume of virus-containing supernatant was added to target cells pre-incubated with 10 ng/L polybrene, and centrifuged at 2500 rpm for 2 hr at 32°C. Following spin inoculation, cells were then incubated overnight at 37°C. Cells expressing the doxycycline-inducible constructs were then selected by sorting mCherry$^{+ve}$ cells. Expression of the HA-tagged designed protein was confirmed with Western blotting using an anti-HA antibody (Roche, Basel, Switzerland; RRID:AB_390918). MEFs were maintained in DME Kelso medium supplemented with 10% (v/v) FBS, 250 mM L-asparagine and 50 mM 2- mercaptoethanol. Colon cancer cell lines were maintained in DMEM/F-12 supplemented with 10% (v/v) FBS.

For constitutive expression of αBCL2, αBCLXL, αBCLW, αMCL1 and αBFL1, genes were first codon optimized for human expression including a 5' Kozak sequence (GCCACC) and 3' FLAG tag, then cloned into the SparQ lentivector containing GFP reporter gene downstream of an internal ribosome entry site (QM530A-1; System Biosciences, Mountain View, CA). Lentiviral particles were produced by transient transfection of 293T cells with plasmid DNA along with packaging constructs pMD2.G and psPAX using calcium chloride precipitation. Viral supernatants were harvested 48 or 72 hr after transfection, filtered and used immediately or stored in aliquots at −80°C.

## MEF cytochrome *c* release assay

Small molecule inhibitors used for cytochrome *c* release and survival assays were purchased from ChemiTek (Indianapolis, IN; ABT-263 and ABT-199) or prepared according to published methods (A-1331852; (*Leverson et al., 2015a*; *Wang et al., 2013*). Mouse embryonic fibroblasts ($1 \times 10^6$) were pelleted and lysed in 0.05% (w/v) digitonin containing lysis buffer (20 mM Hepes-pH 7.2, 100 mM KCl, 5 mM MgCl$_2$, 1 mM EDTA, 1 mM EGTA, 250 mM sucrose), supplemented with protease inhibitors (Roche) for 3 min on ice. Crude lysates containing the mitochondria were incubated with 10 µM ligand at 30°C for 1 hr before pelleting. The supernatant was retained as the soluble fraction (S), while the pellet, containing the mitochondria (P), was solubilized in lysis buffer (20 mM Tris-pH 7.4, 135 mM NaCl, 1.5 mM MgCl$_2$, 1 mM EGTA, 10% (v/v) glycerol and 1% (v/v) Triton X-100. Both soluble and pellet fractions were subsequently analyzed by Western blotting using an anti-cytochrome *c* antibody (BD Biosciences, RRID:AB_396417).

## Short-term survival assays

MEF and colon cancer cells were aliquoted in 96-well tissue culture plates in 50 µL culture media at 20,000 cells per mL. Cells were treated with doxycycline at a final concentration of 1 mg/mL to induce protein expression, and/or small molecule drugs at the indicated final concentrations and a final total volume of 100 µL per well. Viability was assayed after 24 hr with Cell Titer Glo (Promega). For drug titrations, ABT-263 and A-1331852 were serially diluted 2-fold from 250 nM to 2 nM (eight concentrations in total) and combined with doxycycline (to induce expression of αMCL1) or media (drug only). EC$_{50}$ values were determined with nonlinear regression.

HeLa, melanoma, and glioblastoma cell lines (maintained in DMEM with 10% [v/v] FBS) were seeded at 3000–5000 cells per well in 96 well plates in 100 µl culture medium. Cells were transduced the next day with 100 µl lentiviral supernatant to induce expression of each designed inhibitor. For experiments using combinations of three inhibitors, 75 µl media was removed before virus addition to accommodate the appropriate volume of virus. Viability was assayed at 72 hr post-infection with Cell Titer Glo (Promega). Expression of the FLAG-tagged constructs was confirmed by flow cytometry (GFP) and western blotting with an anti-FLAG antibody (Sigma-Alrich, RRID:AB_439685).

## Long-term survival assays

MEF and colon cancers were seeded in 6-well tissue culture plates in 2 mL culture media at 150 cells per mL. The next day and every 48 hr following, doxycycline was added at a final concentration of 1 µg/mL to each well, while nothing was added to control wells. After seven to ten days, media was aspirated and colonies were stained (5:4:1 MeOH:H$_2$O:AcOH, 0.25% Coomassie Blue R-250) and counted.

## Immunoprecipitation

Cells were harvested, washed with PBS, and extracted with ice-cold Chaps buffer (40 mM Tris-HCl, pH 7.5, 150 mM NaCl, 1 mM EDTA, 2% CHAPS, and Complete Protease Inhibitors [Roche]) for 20 min, on ice. Extracts were spun down at 10,000 g for 10 min and supernatants were removed and used for SDS-PAGE analysis. Expression of proteins of interest was analyzed using antibodies against Bcl-2, Bcl-xL, Mcl-1 (as above), Bfl-1 (ProSci,

Inc., Poway, CA; RRID:AB_735550), Bim (BD Biosciences, RRID:AB_397305), and tubulin (Sigma-Aldrich, RRID:AB_477593). For immnoprecipitation experiments, 1000 µg protein lysates were pre-cleared and then incubated with 3 µg Bim antibody for 2 hr at 4°C, followed by addition of Protein A/G Plus agarose beads (Santa Cruz Biotechnology) and overnight incubation with rotation at 4°C. Negative control reactions used normal IgG. Immunoprecipitates were washed four times with lysis buffer and eluted with loading buffer at 95°C, two times for 10 min, followed by SDS-PAGE analysis.

