## [Decision Letter]

Thank you for submitting your article "Computationally designed, high specificity inhibitors delineate the roles of BCL2 family proteins in cancer" for consideration by *eLife*. Your article has been favorably evaluated by Sean Morrison as the Senior Editor and three reviewers, including Brian Kuhlman (Reviewer #3) and Yibing Shan (Reviewer #1), who is a member of our Board of Reviewing Editors.

The reviewers have discussed the reviews with one another and the Reviewing Editor has drafted this decision to help you prepare a revised submission.

Summary:

The manuscript builds upon previous computational design work and reports the design and characterization of a series of selective and high-affinity protein inhibitors of the anti-apoptotic BCl-2 family. This work serves as a good example of how computational design can be used with experimental screening and selection to engineer specific binders against individual members from a family of proteins that share high sequence and structure similarity. The target proteins, pro-survival BCL-2 family proteins, have been the subject of numerous previous studies aimed at designing peptide and small molecule inhibitors. This study presents specific inhibitors for all six homologs and shows how the panel of inhibitors can be used to probe cross-talk and redundancy between BCL-family proteins in human cancer cell lines. The set of probes generated by this work will be very useful for future studies aimed at profiling the roles of BCl-2 family proteins in disease, especially cancers.

While recognizing the quality of the manuscript, the reviewers indicated a number of minor concerns that need to be addressed.

1) In the Abstract there should be a brief mention of the protein engineering strategies that were used to design the specific binders.

2) One interesting thing about the design pipeline is that no explicit negative design (except for the fact that "native" binding residues were kept in some cases) was used during the computational design step, but competition experiments were used at the bench to evolve greater specificity. In hindsight, would a computational technique like multi-state design could have been successfully used to identify mutations that convey specificity (instead of affinity)? For instance, the specific charge interactions that were identified at the edge of the interface. A discussion on this point would be helpful.

3) If one computationally docks the final engineered proteins against the various family members is it possible to pick out which protein they bind to?

4) It is challenging to figure out what is being shown in the various panels of Figure 5. Direct labeling of the panels to indicate what proteins are being shown and whether it is the target interaction or off-target interaction would be helpful. Figure 5 as is too dense. Using two figures to show the current Figure 5 data might be helpful as well.

5) The various control mechanisms outlined in Figure 1 is difficult for non-experts in the area of BCL-2 proteins to follow. For instance, figuring out what the designed inhibitors bind to in the top panel was not straightforward – the dashed line from "designed inhibitors" passes through the activator BOPS and there is a separate dashed line that goes below the pro-survival proteins. It would be better to first present a simpler figure that outlines the design goals, and then in the Discussion present a more complete figure that helps illustrate the complicated set of positive and negative signals.

6) Figure 2: the color scheme selected for the tables makes the dark blue cells illegible.

7) Figure 6—figure supplement 1: It would be informative if the authors also probed for BCl-xL in the BIM co-IP.

---

## [Author Response]

*[…] While recognizing the quality of the manuscript, the reviewers indicated a number of minor concerns that need to be addressed.*

*1) In the Abstract there should be a brief mention of the protein engineering strategies that were used to design the specific binders.*

We have made small modifications to the Abstract.

*2) One interesting thing about the design pipeline is that no explicit negative design (except for the fact that "native" binding residues were kept in some cases) was used during the computational design step, but competition experiments were used at the bench to evolve greater specificity. In hindsight, would a computational technique like multi-state design could have been successfully used to identify mutations that convey specificity (instead of affinity)? For instance, the specific charge interactions that were identified at the edge of the interface. A discussion on this point would be helpful.*

*3) If one computationally docks the final engineered proteins against the various family members is it possible to pick out which protein they bind to?*

To address comments 2 and 3, we have computationally docked each designed protein and optimized variant into the binding grooves of each BCL2 pro-survival homolog (Figure 5—figure supplement 2 and Figure 5—figure supplement 3). Overall, both the partially-specific computational designs and very specific optimized inhibitors exhibit more favorable absolute binding energy (minimum ddG of local docking trajectories) and relative binding energy (minimum ddG of local docking compared to minimum of global docking) when docked to on-target homologs compared to off-target homologs. These calculations resemble trends in the experimental binding data, but they do not discriminate between the highly specific, optimized inhibitors and partially specific precursors. Thus, while additions to the design protocol such as computational docking or multi-state design against off-target homologs may improve the initial success rate of achieving high affinity and at least partially specific binding, the resolution of these calculations limits discrimination between variants with low versus high specificity. These data and accompanying explanation have been added to the manuscript.

*4) It is challenging to figure out what is being shown in the various panels of Figure 5. Direct labeling of the panels to indicate what proteins are being shown and whether it is the target interaction or off-target interaction would be helpful. Figure 5 as is too dense. Using two figures to show the current Figure 5 data might be helpful as well.*

We have improved labeling and modified the figure legend for clarity, and split the figure in two (now Figure 5 and Figure 6) as recommended.

*5) The various control mechanisms outlined in Figure 1 is difficult for non-experts in the area of BCL-2 proteins to follow. For instance, figuring out what the designed inhibitors bind to in the top panel was not straightforward – the dashed line from "designed inhibitors" passes through the activator BOPS and there is a separate dashed line that goes below the pro-survival proteins. It would be better to first present a simpler figure that outlines the design goals, and then in the Discussion present a more complete figure that helps illustrate the complicated set of positive and negative signals.*

We think the schematic is appropriate to include in the Introduction, in which we describe the biological significance of BCL2 inter-family interactions and summarize current BH3-mimetic inhibitors. Per suggestion, we have modified this figure for clarity, and added an additional figure summarizing the design strategy (Figure 1—figure supplement 1).

*6) Figure 2: the color scheme selected for the tables makes the dark blue cells illegible.*

We have modified the color scheme.

7) Figure 6—figure supplement 1: It would be informative if the authors also probed for BCl-xL in the BIM co-IP.

We have modified the figure (now Figure 7—figure supplement 1) to include this data.